# Accelerating Spectral Clustering under Fairness Constraints

**Francesco Tonin** [1] **Alex Lambert** [2] **Johan A.K. Suykens** [2] **Volkan Cevher** [1]

## Abstract

Fairness of decision-making algorithms is an increasingly important issue. In this paper, we focus on spectral clustering with group fairness constraints, where every demographic group is represented in each cluster proportionally as in the general population. We present a new efficient method for fair spectral clustering (Fair SC) by casting the Fair SC problem within the difference of convex functions (DC) framework. To this end, we introduce a novel variable augmentation strategy and employ an alternating direction method of multipliers type of algorithm adapted to DC problems. We show that each associated subproblem can be solved efficiently, resulting in higher computational efficiency compared to prior work, which required a computationally expensive eigendecomposition. Numerical experiments demonstrate the effectiveness of our approach on both synthetic and real-world benchmarks, showing significant speedups in computation time over prior art, especially as the problem size grows. This work thus represents a considerable step forward towards the adoption of fair clustering in real-world applications.

## 1. Introduction

Algorithmic decision-making systems leveraging machine learning (ML) are increasingly being used in critical domains such as healthcare, social policy, and education, raising concerns about the potential for these algorithms to exhibit unfair behavior towards certain demographic groups (Hardt et al., 2016; Buolamwini & Gebru, 2018; Chouldechova & Roth, 2020). In response to these concerns, the field of fair ML has proposed mathematical fairness formulations for various ML tasks, e.g., (Dwork et al., 2012; Zafar et al., 2017; Samadi et al., 2018; Donini et al., 2018;

Agarwal et al., 2019; Aghaei et al., 2019; Amini et al., 2019; Davidson & Ravi, 2020; Celis et al., 2018; Singh et al., 2023; Ali et al., 2023).

In clustering research, Chierichetti et al. (2017) introduced demographic fairness by imposing fairness constraints to ensure balanced representation across protected groups in clusters. Initially applied to two groups in Chierichetti et al. (2017), this concept was expanded to multiple groups (Rösner & Schmidt, 2018; Bera et al., 2019a) and primarily explored in prototype-based clustering (Chierichetti et al., 2017; Carreira-Perpinán & Wang, 2013; Bera et al., 2019a). Kleindessner et al. (2019) adapted this notion of fairness to spectral clustering (Shi & Malik, 2000; Von Luxburg, 2007) and is known as fair spectral clustering (Fair SC). Although recent advances (Wang et al., 2023) have sped up the computation of Fair SC, the reliance on the computationally expensive eigendecomposition of the fairness-constrained graph Laplacian still limits the application of Fair SC to real-world problems.

From an optimization standpoint, SC can be cast as a trace maximization problem with orthonormality constraints (Bach & Jordan, 2003), falling into the differences of convex functions (DC) framework. This problem class has spurred substantial interest (Tao et al., 1986; Le Thi & Pham Dinh, 2018), and efficient DC-based algorithms have been developed for various ML tasks including feature selection (Le Thi et al., 2015), reinforcement learning (Piot et al., 2014) and (kernel) PCA (Beck & Teboulle, 2021; Tonin et al., 2023). However, these algorithms do not extend directly to Fair SC due to their lack of consideration for the fairness constraints, the integration of which within the DC framework remains unexplored in the existing literature.

In this work, we cast the Fair SC problem into the DC framework and develop an efficient alternating direction method-of-multipliers (ADMM)-type algorithm through a suitable variable augmentation, achieving higher computational efficiency over existing algorithms (Kleindessner et al., 2019; Wang et al., 2023). Our specific contributions can be summarized as follows:

- *Novel Algorithm Design*: We develop a new efficient optimization method for Fair SC by casting the problem in the DC framework and by designing a novel

---

[1]LIONS, EPFL, Switzerland [2]ESAT-STADIUS, KU Leuven, Belgium. Correspondence to: Francesco Tonin <francesco.tonin@epfl.ch>.

*Proceedings of the 42nd International Conference on Machine Learning*, Vancouver, Canada. PMLR 267, 2025. Copyright 2025 by the author(s).

variable augmentation suitable for efficient DC optimization within an ADMM type of algorithm.

- *Efficient Solution via DC*: We show that each associated ADMM subproblem can be solved efficiently. In particular, our approach can exploit fast gradient-based algorithms for the DC framework. This results in better computational efficiency of our method, avoiding the expensive eigendecomposition of a modified Laplacian required by existing algorithms.

- *Empirical Validation*: Through numerical experiments, we demonstrate the effectiveness of our approach on both synthetic and real-world benchmarks, showing significant speedups in computation time, especially with larger sample size and number of clusters.

This paper is structured as follows. Section 2 reviews the group fairness constraints for spectral clustering and the existing algorithms for Fair SC. Section 3 presents our new algorithm for Fair SC. The numerical experiments in Section 4 show the advantage of our method in computational efficiency over existing algorithms on multiple benchmarks. Proofs are deferred to the Appendix.

## 2. Problem Formulation

**Notation:** Given a symmetric real matrix $M \in \mathbb{R}^{n \times n}$, $\lambda(M) \in \mathbb{R}^n$ is the vector of its eigenvalues ordered decreasingly. $I_s$ is the identity matrix of size $s \times s$. $\|\cdot\|_{\mathrm{F}}$ denotes the Frobenius norm. For a convex set $\mathcal{C}$, $\iota_{\mathcal{C}}(\cdot)$ is its indicator function: 0 on $\mathcal{C}$ and $+\infty$ otherwise. The Fenchel-Legendre transform of a function $f$ is $f^\star$. For an integer $s > 0$, $[s]$ denotes the set $\{1, \ldots, s\}$.

### 2.1. Group-fair clustering

Given a set of data points, the goal of clustering is to partition the dataset into disjoint subsets such that data points in the same subset are more similar to each other than to those in the other subsets. Formally, let $\mathcal{D} = \{x_i \in \mathbb{R}^d\}_{i=1}^n$ be a dataset of $n$ data points. Clustering partitions $\mathcal{D}$ into $k$ clusters:

$$\mathcal{D} = C_1 \cup \cdots \cup C_k, \tag{1}$$

such that the resulting clustering has high intra-cluster similarity and low inter-cluster similarity. Clustering can be encoded in a clustering indicator matrix $Q \in \mathbb{R}^{n \times k}$, where $q_{il} := 1$ if $x_i \in C_l$, and 0 otherwise, for $i \in [n]$ and $l \in [k]$. We now review the notion of group fairness in clustering. Suppose we are also given $h$ groups partitioning the dataset $\mathcal{D}$ (e.g., based on sensitive data such as nationality or census):

$$\mathcal{D} = V_1 \cup \ldots \cup V_h, \tag{2}$$

where $V_i \cap V_j = \emptyset$ for $i \neq j$. Chierichetti et al. (2017); Bera et al. (2019a) proposed the following notion of balance

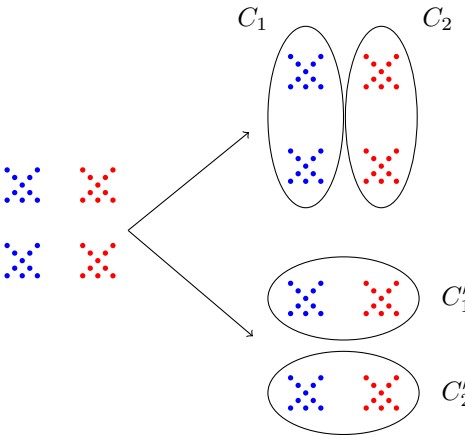

Figure 1: Illustrative example of fair clustering. Red and blue colors indicate $h = 2$ different demographic groups. Clustering with $k = 2$ of the left-hand side data can result in two possible partitionings: the top clustering $C_1, C_2$ or the bottom clustering $C_1', C_2'$, where balance$(C_{\{1,2\}}) = 0$ and balance$(C'_{\{1,2\}}) = 1$.

in fair clustering such that each cluster contains the same number of elements from each group $V_s$.

**Definition 2.1** (Balance (Chierichetti et al., 2017))**.** For a clustering of type (1), define the balance of the cluster $C_l$ as

$$\text{balance}(C_l) = \min_{\substack{s, s' \in [h] \\ s \neq s'}} \frac{|V_s \cap C_l|}{|V_{s'} \cap C_l|} \in [0, 1]. \tag{3}$$

An example of fair clustering is shown in Figure 1, with ground-truth labels shown for visualization purposes. In group-level fairness (Chierichetti et al., 2017), the higher the balance of each cluster, the fairer the clustering. A perfectly balanced clustering means that objects from all groups are presented proportionately in each cluster. The following definition due to Kleindessner et al. (2019) gives the corresponding group fairness condition.

**Definition 2.2** (Group fairness)**.** A clustering such as (1) is group fair with respect to a group partition (2) if the proportion of each group in all clusters is the same as in $\mathcal{D}$, i.e. for $s \in [h]$ and $l \in [k]$,

$$\frac{|V_s \cap C_l|}{|C_l|} = \frac{|V_s|}{n}. \tag{4}$$

Groups can be encoded in a group indicator matrix $G \in \mathbb{R}^{n \times h}$ with $g_{is} := 1$ if $x_i \in V_s$, and 0 otherwise, for $i \in [n]$, $s \in [h]$. The fairness condition (4) can be represented in matrix form using the indicator matrices $Q$ and $G$. This is done in Wang et al. (2023) by considering the matrices $A = G^\top Q$ and $B = (G^\top 1_n) \cdot (Q^\top 1_n)^\top$. The entries of $A$ and $B$ are $a_{sl} = |V_s \cap C_l|$ and $b_{sl} = |V_s| \cdot |C_l|$ for

$s \in [h]$ and $l \in [k]$. The group-fairness condition (4) is then equivalent to $n \cdot A = B$, or

$$F_0^\top Q = 0, \tag{5}$$

where $F_0 = G - 1_n z^T \in \mathbb{R}^{n \times h}$ and $z = \frac{G^T 1_n}{n} \in \mathbb{R}^h$.

## 2.2. Fair spectral clustering

Given the dataset $\mathcal{D}$, in this work we consider a positive definite kernel $\kappa : \mathbb{R}^d \times \mathbb{R}^d \to \mathbb{R}$ and define the affinity matrix $K \in \mathbb{R}^{n \times n}$ as $K_{ij} = \kappa(x_i, x_j)$ and the degree matrix $D \in \mathbb{R}^{n \times n}$ as $D_{ii} = \sum_{j=1}^n K_{ij}$ (Alzate & Suykens, 2008). The spectral clustering solution is obtained by solving an eigenvalue problem of size $n \times n$ and corresponds to the eigenvectors $H \in \mathbb{R}^{n \times k}$ associated with the $k$ largest eigenvalues of the normalized affinity matrix $M := D^{-1/2} K D^{-1/2}$. Typically as in normalized SC, the clustering indicators are then obtained by applying $k$-means to the rows of $D^{-1/2} H$.

*Remark* 2.3. Spectral clustering with kernel as affinity function is particularly useful in clustering applications where the input data is given as a data matrix. Our algorithm can also be applied in the standard SC setting (Shi & Malik, 2000; Von Luxburg, 2007) where the input is given as a graph with adjacency matrix $W$, by considering $\mathcal{M} = D^{-1/2} W D^{-1/2}$. We can regularize the problem by working with p.s.d. $M := \mathcal{M} + (1 + \omega) I_n$ for small $\omega > 0$. In fact, it holds that $\lambda_i(\mathcal{M}) \geq -1, \forall i = 1, \dots, n$, as the eigenvalues of the normalized Laplacian $\hat{L} = D^{-1/2} L D^{-1/2}$, with $L = D - W$, lie in $[0, 2]$. This regularization does not alter the solution space as it only shifts the eigenvalues. This is a well-studied regularization technique that is widely used, e.g., in kernel methods (Bishop & Nasrabadi, 2006).

The group fairness constraint was incorporated into the framework in Kleindessner et al. (2019) by adding the linear constraint (5) to the spectral clustering problem:

$$\begin{aligned} \max_{H \in \mathbb{R}^{n \times k}} \quad & \text{Tr}(H^\top M H) \\ \text{s.t.} \quad & H^\top H = I_k, \\ & F^\top H = 0 \end{aligned} \tag{6}$$

with normalization $F = D^{-1/2} F_0$, as reviewed in Appendix D.2. We now review existing algorithms for (6).

### 2.2.1. O-FSC ALGORITHM

The nullspace-based algorithm for solving (6) proposed in Kleindessner et al. (2019) is as follows. Since the columns of $H$ live in the null space of $F^\top$, we can write $H = ZY$, for some $Y \in \mathbb{R}^{(n-h) \times k}$, where $Z \in \mathbb{R}^{n \times (n-h)}$ is an orthonormal basis of $\text{null}(F^\top)$. Consequently, the optimization problem (6) is equivalent to the following trace maximization where the linear constraints are removed:

$$\max_{Y \in \mathbb{R}^{(n-h) \times k}} \text{Tr}\left(Y^\top M_Z Y\right) \quad \text{s.t.} \quad Y^\top Y = I_k, \tag{7}$$

with modified affinity $M_Z = Z^\top M Z \in \mathbb{R}^{(n-h) \times (n-h)}$. By Bach & Jordan (2003), the solution $Y$ is given by the $k$ largest eigenvectors of $M_Z$ by the eigenvalue problem

$$M_Z Y = Y \Lambda, \tag{8}$$

where $\Lambda = \text{diag}(\lambda_{n-h-k}, \dots, \lambda_{n-h})$ is the diagonal matrix of eigenvalues of $M_Z$. Finally, the solution to the original problem (6) is recovered as $H = D^{-1/2} ZY$.

This algorithm is denoted o-FSC as it is the original algorithm for Fair SC. o-FSC requires two major computational steps. The first one is the explicit computation of the null space of $F^T$. This can be done by the SVD of $F$, which has time complexity $\mathcal{O}(nh^2)$. The second major computational step is the eigenvalue decomposition of $M_Z$. This step has complexity $\mathcal{O}((n-h)^3)$. Due to the cubic complexity of eigendecomposition, the o-FSC algorithm is only suitable for small $n$ and is not scalable to real-world datasets.

### 2.2.2. S-FSC ALGORITHM

In Wang et al. (2023), the authors proposed a more efficient version of the o-FSC algorithm. Wang et al. (2023) rewrite the eigenvalue problem (8) in terms of a new matrix s.t. they can efficiently apply the the implicitly restarted Arnoldi method (Sorensen, 1992) for computing eigenpairs. First note that (8) can be rewritten by left-multiplication with $Z$ as

$$\left(ZZ^\top M ZZ^\top\right) ZY = ZY \Lambda, \tag{9}$$

as $Z^\top Z = I_{n-h}$. This leads to the following projected eigenvalue problem $AY' = Y' \Lambda$, where $A = PMP \in \mathbb{R}^{n \times n}$ with $P = ZZ^\top$ and $Y' = ZY$. It is possible to show that an eigenvalue/eigenvector pair $(\lambda, y')$ of $A$, with $\lambda$ being one of its largest $n - h$ eigenvalues, is also an eigenvalue/eigenvector pair of $M_Z$ with $y = Z^\top y'$. Therefore, one can solve Problem (8) by finding the smallest eigenvectors of (9). Note that, for the sake of avoiding the computation of $Z$, Wang et al. (2023) use the rows of $H' = D^{-1/2} Y'$ instead of $H = D^{-1/2} ZY'$ in the $k$-means step, with $Y'$ being the top $k$ eigenvectors of $A$. The complexity of s-FSC for each eigensolver iteration is $\mathcal{O}(n^2 + nh^2 + nk^2)$, with constants depending on the number of required restarts of the Arnoldi method; in general, this depends on the initial vector and properties of $M$, in particular the distribution of its eigenvalues (Stewart, 2001). While the s-FSC algorithm improves efficiency over o-FSC, its scalability remains limited by the eigendecomposition routine, which in practice exhibits significant computational cost.

## 3. Algorithm

In this section, we design an ADMM-like algorithm for Fair SC. This is done by reformulating the problem within the difference of convex functions (DC) framework and devising a

new variable augmentation allowing efficient dualization of the ADMM subproblem. Our specific construction leads to efficient gradient-based algorithms avoiding the expensive eigendecomposition routines on the $n \times n$ matrices.

**Difference of convex functions.** To leverage the efficiency of DC optimization, we recast Problem 6 as the minimization of a difference of convex functions. However, directly applying DC optimization to Problem 6 would necessitate computing $M^{1/2}$, a computationally demanding operation akin to solving the original problem. To circumvent this challenge, we formulate our DC problem in terms of $M^2$ instead of $M$. This choice allows us to bypass the computation of $M^{1/2}$ without sacrificing solution quality, as we will demonstrate empirically that optimizing with $M^2$ yields solutions exhibiting comparable fairness to those obtained with $M$. Note that in practice we never compute $M^2$ explicitly, as we only need to compute matrix-vector products. Formally, we define $f, g, h \colon \mathbb{R}^{n \times k} \to \mathbb{R} \cup \{+\infty\}$ as follows:

$$f(H) = \frac{1}{2} \left\| H \right\|_{\mathrm{F}}^2, \qquad g(H) = \iota_{\mathcal{S}_n^k}(H),$$
$$h(H) = \iota_{\{0\}}(F^\top H).$$

*Remark* 3.1. Here, the orthogonality constraints correspond to belonging to the Stiefel manifold $\mathcal{S}_n^k = \{H \in \mathbb{R}^{n \times k} \mid H^\top H = I_k\}$. While the constraint $H \in \mathcal{S}_n^k$ is not itself convex, it can be relaxed into a convex constraint by considering the convex hull of the Stiefel manifold as the solutions necessarily lie on the boundary (Uschmajew, 2010, Lemma 2.7).

Using this notation, our proposed DC formulation reads

$$\min_{H \in \mathbb{R}^{n \times k}} g(H) + h(H) - f(MH). \qquad (10)$$

Note that Problem 10 corresponds to solving a modified (6) with $M^2$ instead of $M$ in the cost function. Notice that, without the fairness constraints, (10) reduces to finding the top $k$ eigenvectors of $M^2$ (Bach & Jordan, 2003). While efficient optimization algorithms with DC for the simpler variance-maximization case without fairness constraints (i.e., $h = 0$) have been studied in previous work (e.g., in the PCA literature (Thiao et al., 2010; Beck & Teboulle, 2021; Tonin et al., 2023)), the main **challenge** here is to develop an efficient algorithm that can handle the additional complexity introduced by the fairness constraints. In this work, we derive a dual framework that is amenable to fast gradient-based optimization of (10), as opposed to the eigendecomposition routines used in o-FSC (Kleindessner et al., 2019) and s-FSC (Wang et al., 2023).

**Novel variable augmentation.** To address the Fair SC problem, we propose to first cast the problem into an

ADMM framework. The ADMM approach in the context of difference of convex functions has been investigated in, e.g., (Sun et al., 2018a; Chuang et al., 2022; Tu et al., 2020). However, naively applying the existing ADMM algorithms to the Fair SC problem results in intermediate $\arg\min$ problems that are not easy to solve, involving large matrix inversions. To arrive at efficient solutions, we introduce a novel variable augmentation scheme, where the linear constraint couples the ADMM variable $Y$ with $MH$ instead of directly with $H$. This specific design choice is crucial for decomposing the problem into a subproblem w.r.t. $H$ that can be efficiently tackled through dualization within the DC framework. Moreover, as we will demonstrate empirically, enforcing fairness on $MH$ instead of $H$ effectively promotes the same group balance. Let us then write (10) with the proposed variable augmentation as a composite function with linear constraints:

$$\min_{H, Y \in \mathbb{R}^{n \times k}} g(H) + h(Y) - f(MH) \quad \text{s.t.} \quad MH = Y. \quad (11)$$

We form the augmented Lagrangian

$$\begin{aligned} \mathcal{L}(H, Y, P) = {} & g(H) + h(Y) - f(MH) \\ & + \langle P, MH - Y \rangle \\ & + \frac{\alpha}{2} \left\| MH - Y \right\|_{\mathrm{F}}^2, \end{aligned} \qquad (12)$$

where $P$ are the Lagrange multipliers and $\alpha$ is the penalty parameter. The ADMM algorithm corresponds to the following iterations:

$$H^{(i+1)} = \arg\min_{H \in \mathbb{R}^{n \times k}} \mathcal{L}(H, Y^{(i)}, P^{(i)}), \qquad (13)$$

$$Y^{(i+1)} = \arg\min_{Y \in \mathbb{R}^{n \times k}} \mathcal{L}(H^{(i+1)}, Y, P^{(i)}), \qquad (14)$$

$$P^{(i+1)} = P^{(i)} + \alpha(MH^{(i+1)} - Y^{(i+1)}). \qquad (15)$$

Our algorithm is summarized in Algorithm 1. The penalty $\alpha^{(i+1)}$ is updated according to the standard rule suggested in (Boyd et al., 2011) and detailed in Appendix B. We now analyze the two subproblems separately.

### 3.1. Subproblem with respect to $H$

Our approach to solving Problem (13) consists in the following steps. First, we remark that it can be written as a difference of convex functions. Such problems have been widely studied in the literature (Tao et al., 1986), and in our case dualization is possible and strong duality holds. Next, we solve the dual problem using iterative techniques, as we can provide an explicit form for the corresponding gradient using standard properties of the Fenchel-Legendre conjugates. Finally, we can get back the primal solution that is exactly the solution to Problem (13) by computing the SVD of a well-chosen matrix.

**Algorithm 1** Proposed ADMM-type method for Fair SC

---

**Input:** affinity matrix $K \in \mathbb{R}^{n \times n}$; group matrix $F \in \mathbb{R}^{n \times h}$; $k \in \mathbb{N}$; $\alpha^{(0)} > 0$

1: Compute normalized affinity matrix $M = D^{-1/2} K D^{-1/2}$
2: Initialization $H^{(0)} = 0, Y^{(0)} = 0, P^{(0)} = 0$
3: $\mathcal{L}_\alpha(H, Y, P) = g(H) + h(Y) - f(MH) + \langle P, MH - Y \rangle + \frac{\alpha}{2} \|MH - Y\|_F^2$
4: **for** $i = 0, 1, \ldots, T-1$ **do**
5: $\quad H^{(i+1)} = \arg\min_H \mathcal{L}_{\alpha^{(i)}}(H, Y^{(i)}, P^{(i)})$
6: $\quad Y^{(i+1)} = \arg\min_Y \mathcal{L}_{\alpha^{(i)}}(H^{(i+1)}, Y, P^{(i)})$
7: $\quad P^{(i+1)} = P^{(i)} + \alpha^{(i)} (MH^{(i+1)} - Y^{(i+1)})$
8: $\quad$ Update $\alpha^{(i+1)}$ according to (19)
9: **end for**
10: Apply $k$-means clustering to the rows of $H' = D^{-\frac{1}{2}} H^{(T)}$

---

**Proposition 3.2.** *Let $\phi \colon X \mapsto f(X) - \langle P, X \rangle - \frac{\alpha}{2} \|X - Y\|^2$ and assume that $\alpha < 1$. Then Problem (13) can be written as a DC. Moreover, it holds that*

$$\arg\min_{H \in \mathbb{R}^{n \times k}} \mathcal{L}(H, Y^{(i)}, P^{(i)}) = \arg\min_{H \in \mathbb{R}^{n \times k}} g(H) - \phi(MH)$$

*and the dual problem reads*

$$\inf_{V \in \mathbb{R}^{n \times k}} \phi^\star(V) - g^\star(MV). \tag{16}$$

*Finally, strong duality holds.*

The condition $\alpha < 1$ is not a limiting assumption as the ADMM is known to converge for small $\alpha$ values (Nocedal & Wright, 1999). To solve (16), we assume that $M$ is full rank, which typically happens when the affinity matrix $K$ comes from positive definite kernels associated with infinite dimensional feature spaces, such as the Gaussian kernel, and the data dataset does not contain any duplicates.

This assumption is needed to guarantee the existence of the gradients of the terms of Problem (16) that are detailed in the following proposition.

**Proposition 3.3.** *Let $V \in \mathbb{R}^{n \times k}$, and $A \colon V \mapsto \frac{1}{1-\alpha}(V + P^{(i)} - \alpha Y^{(i)})$. Then $\phi^\star(V) = \frac{1}{2} \|V\|^2 - \frac{1}{2} \|A(V) - V\|^2 + \frac{\alpha}{2} \|A(V) - Y^{(i)}\|^2 + \langle P^{(i)}, A(V) \rangle$ and $g^\star(MV) = \mathrm{Tr}\left(\sqrt{V^\top M^2 V}\right)$.*

**Gradient-based techniques for the dual problem.** Solving Problem (16) can be done by using iterative gradient-based techniques, as all the quantities involved are easy to compute. Indeed, $\phi^\star$ is a quadratic form whose gradient is trivial and, for $g^\star(MV)$, we exploit the fact that its gradient is known in closed-form (Tonin et al., 2023) and expressed through the SVD of the matrix $V^\top M^2 V \in \mathbb{R}^{k \times k}$; this SVD

is computationally cheap in the context of a relatively small number of clusters. Overall, the computational complexity per iteration associated to the computation of a dual solution $\hat{V}$ is bounded by the sum of: (i) the computation of $V^\top M^2 V$ that scales as $\mathcal{O}(kn^2)$ (ii) the SVD of $V^\top M^2 V$ that costs $\mathcal{O}(k^3)$ (iii) the matrix products for $\nabla g^\star(MV)$ in $\mathcal{O}(nk^2 + k^3)$ (iv) the computation of $\nabla\phi^\star$ that scales as $\mathcal{O}(nk)$. In the experiments, we solve Problem (16) using a fast L-BFGS optimization algorithm. Note that $M^2$ is not explicitly computed as one can use the associative property to perform consecutive multiplications, e.g. $V^\top M^2 V$ can be computed as $(V^\top M)(MV)$.

**Computing the primal solution.** Once the dual solution $\hat{V}$ is computed, one can recover the corresponding primal solution $H^{(i+1)}$ by

$$\max_{H \in \mathbb{R}^{n \times k}} \langle \hat{V}, MH \rangle \quad \text{s.t.} \quad H^\top H = I_k. \tag{17}$$

This step can be performed by $H^{(i+1)} = LR^\top$ computing the SVD of the matrix $\hat{V}^\top M = LSR^\top$, with complexity $\mathcal{O}(nk^2)$.

### 3.2. Subproblem with respect to $Y$

We now turn to solving Problem (14), i.e.

$$\arg\min_{Y \in \mathbb{R}^{n \times k}} \frac{\alpha}{2} \left\|MH^{(i+1)} - Y\right\|_F^2 - \langle P^{(i)}, Y \rangle$$
$$\text{s.t.} \quad F^\top Y = 0. \tag{18}$$

The linear constraints can be removed by using a suitable parameterization $Y = QZ$ where $Q \in \mathbb{R}^{n \times (n-h)}$ is the matrix of an orthonormal basis of the nullspace of $F^\top$ that can be obtained from the SVD of $F$ similarly to the technique employed in Section 2.2.1. Then $Z \in \mathbb{R}^{(n-h) \times k}$ is the new variable, resulting in the problem in $Z$ without linear constraints. The problem is solved by nullifying the gradient in closed-form solution, i.e., $\hat{Z} = Q^\top MH^{(i+1)} + \frac{1}{\alpha} Q^\top P^{(i)}$ and $Y^{(i+1)} = Q\hat{Z}$. The computational complexity in this step is dominated by matrix-vector products in $\mathcal{O}(kn^2)$ and by the SVD of $F$ in $\mathcal{O}(nh^2)$. We note that this SVD is in practice fast as $h$ is typically very small.

**Complexity analysis.** The complete ADMM-like algorithm is summarized in Algorithm 1. The runtime is dominated by matrix multiplication applying $M$ to an $n \times k$ matrix (i.e., the $H$ variable and its dual $V$), with complexity $\mathcal{O}(n^2 k)$, where typically $k \ll n$. The improvement is therefore in the efficiency of the core operations. In fact, while for an $n \times n$ matrix both matrix multiplication and matrix eigendecomposition have the same $\mathcal{O}(n^3)$ complexity, the former is much more efficient in practice. The proposed efficient gradient-based algorithm avoids the eigendecomposition routines and is shown to achieve significantly higher computational efficiency in practice in Section 4.

Table 1: **m-SBM.** Runtime and balance for m-SBM benchmark with $k = 50, h = 5$ and varying $n$.

| $n$ | Metric | o-FSC | s-FSC | Ours |
|---|---|---|---|---|
| 5000 | Time (s) | 96.39 | 75.61 | 3.29 |
| | Balance | 1.00 | 1.00 | 1.00 |
| 7500 | Time (s) | 235.62 | 88.99 | 5.29 |
| | Balance | 1.0 | 1.0 | 1.0 |
| 10000 | Time (s) | 495.70 | 107.94 | 9.19 |
| | Balance | 1.0 | 1.0 | 1.0 |

**Convergence analysis.** While ADMM convergence is well-established for convex problems (Eckstein & Bertsekas, 1992; Boyd et al., 2011), obtaining convergence results for nonconvex problems is an active area of research. Existing results rely on problem-specific assumptions such as in the nonconvex consensus setting (Hong et al., 2016), or require structural properties such as bounded Hessians (Li & Pong, 2015) or continuity (Wang et al., 2019) preventing their direct application to Problem 6. Other works (Sun et al., 2018b; Pham et al., 2024) exploit the difference of convex functions structure but are limited to differentiable convex functions $h$ with Lipschitz continuous gradient. Finally, another line of works rely on making additional assumptions on the sequence of iterates (Shen et al., 2014; Jiang et al., 2014; Magnusson et al., 2016), yielding weaker results. In particular, applying Proposition 3 from Magnusson et al. (2016) to (11), we characterize the convergence of the ADMM algorithm as specified in the following proposition, whose formal details can be found in Appendix D.1.

**Proposition 3.4.** *Provided that the sequence of dual variables $P^{(i)}$ generated by Algorithm 1 converges, any limit point $(H^*, Y^*)$ of the primal sequence $(H^{(i)}, Y^{(i)})$ satisfies the first-order conditions.*

The proof of Proposition 3.4 relies on (i) Problem (11) having finitely defined closed constraints sets, (ii) local solutions for subproblems, and (iii) limit solutions enjoying a regular set of constraint gradient vectors. Beyond theoretical guarantees, ADMM algorithms have been successfully applied to a wide array of nonconvex problems in machine learning, e.g., (Xu et al., 2012; Sun & Fevotte, 2014). Empirical evidence on a wide range of tasks for both synthetic and real-world datasets presented in the next section demonstrates the strong practical performances of Algorithm 1.

## 4. Numerical Experiments

Through numerical evaluations on both synthetic and real-world datasets, we show the efficiency of the proposed ADMM-type algorithm with DC dualization for Fair SC. We compare with the original Fair SC algorithm (o-FSC)

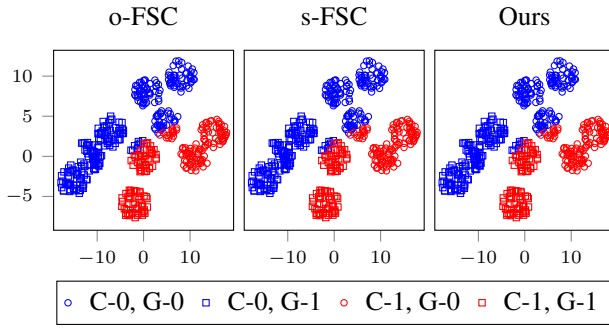

Figure 2: **Fair clustering of Elliptical dataset**. The clustering label is represented by different colors, and sensitive attributes by shapes. The legend is "C-$i$, G-$j$" for Cluster-$i$, Group-$j$. These plots show that our method produces assignments comparable to exact algorithms (o-FSC, s-FSC). Critically, we achieve this with reduced computations.

Table 2: **Real-world data**. Runtime for multiple Fair SC real-world problems with $k = 25$.

| Dataset | $n$ | Time (s) | | |
|---|---|---|---|---|
| | | o-FSC | s-FSC | Ours |
| LastFMNet | 5576 | 103.82 | 19.08 | **4.59** |
| Thyroid | 7200 | 279.03 | 30.49 | **7.38** |
| Census | 32561 | - | 136.60 | **15.78** |
| 4area | 35385 | - | 166.92 | **25.85** |

(Kleindessner et al., 2019) and its state-of-the-art scalable extension (s-FSC) (Wang et al., 2023). We apply the proposed algorithm to problems of different sizes, quantify the fairness of the computed clusters, and compare solutions with exact methods. We also study the effect of the number of clusters $k$ on runtime and conduct a sensitivity analysis on the penalty parameter $\alpha$. Experiments are implemented in Python 3.10 on a machine with a 3.6GHz Intel i7-9700K processor and 64GB RAM.

**Datasets.** We consider the following synthetic datasets: m-SBM, RandLaplace, and Elliptical. The m-SBM benchmark (Kleindessner et al., 2019) is a modification of the stochastic block model (Holland et al., 1983) to take fairness into account. It has ground-truth perfectly fair clustering, where the $n$ nodes are partitioned in $h$ groups and assigned to a fixed fair clustering with $k$ clusters. In the experiments, we vary $n$ and set $k = 50, h = 5$ and edge connectivity probability as $(\frac{\log n}{n})^{1/10}$. The RandLaplace dataset is a graph induced by a random $n \times n$ symmetric adjacency matrix $W$ with each node randomly assigned to one of $h = 2$ groups. Elliptical has $k = 2$ clusters and $h = 2$ groups, as defined in (Feng et al., 2024). We consider the following real-world datasets, summarized in Table 10 in Appendix. LastFMNet

(Rozemberczki & Sarkar, 2020) ($n = 5576, h = 6$) is the graph of follower relationship between users of the Last.fm website; the groups correspond to different nationalities. Thyroid (Quinlan, 1987) ($n = 7200, h = 3$) contains 21 attributes of patients divided into three groups: not hypothyroid, hyperfunction, and subnormal functioning. Census (Kohavi et al., 1996) ($n = 32561, h = 7$) contains 12 attributes from 1994 US census data, with 7 groups representing demographic categories. 4area (Ahmadian et al., 2019) ($n = 35385, h = 4$) represents researchers from four areas of computer science: data mining, machine learning, databases, and information retrieval. The RBF kernel is used for Thyroid, Census, and 4area, while the remaining datasets are given as graphs through Remark 2.3.

**Metrics.** The average balance, as introduced in Chierichetti et al. (2017), is used to measure how fair a clustering is; it is the average over all clusters of (3). It is a value in $[0, 1]$, where a value of 1 corresponds to a perfectly fair clustering, i.e., (4) holds. Note that our approach follows the established line of works on fair spectral clustering, which do not provide guarantees in the general case on the balance of the clustering given by (6), as discussed in Kleindessner et al. (2019), similarly to how the relaxed spectral clustering only approximates the optimal normalized cut solution (Shi & Malik, 2000). All reported running times are averaged over 5 trials. Additional results and detailed setups are provided in Appendix A and B.

### 4.1. Experimental results

**Synthetic benchmarks.** In this experiment, we compute the Fair SC solution on the m-SBM benchmark (Kleindessner et al., 2019) to evaluate the performance and efficiency of our proposed method. The results are presented in Table 1. The experiment is carried out for sample sizes $n \in \{5000, 7500, 10000\}$. Our method consistently outperforms both o-FSC and s-FSC in runtime. For instance, at $n = 10000$, our method is approximately 54 times faster than o-FSC and 12 times faster than s-FSC. In terms of average balance, recall that Problem 6 is able to recover the ground-truth fair clustering in m-SBM (Kleindessner et al., 2019). This is confirmed in the experiments, where the balance is consistently 1.0 for all three methods across sample sizes, demonstrating that all methods equally respect the fairness constraints. We also consider the Elliptical dataset (Feng et al., 2024) ($k = 2, h = 2$) and visualize the found clustering in Figure 2, showing that our method achieves comparable labels to the exact algorithms.

**Real-world data.** We now test on real-world datasets to compare the performance of our proposed method with o-FSC (Kleindessner et al., 2019) and s-FSC (Wang et al., 2023). The runtime of the algorithms was measured for a

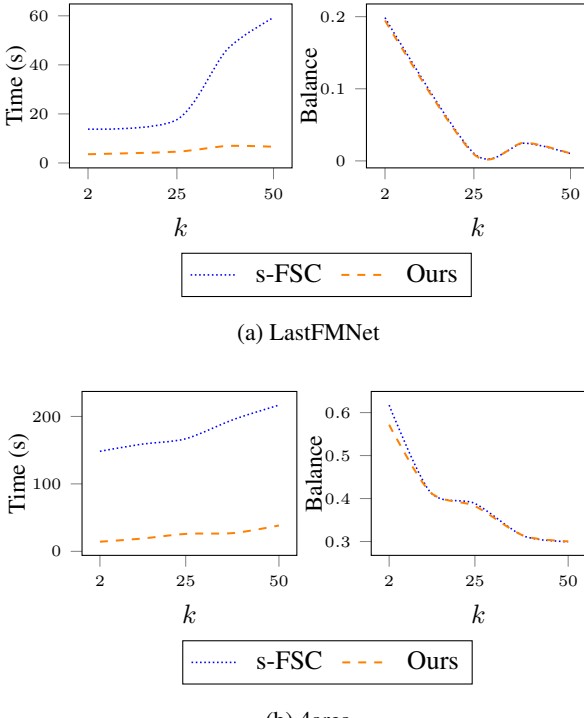

(a) LastFMNet

(b) 4area

Figure 3: **Runtime and fairness across $k$.** Fair SC on real-word datasets (a) LastFMNet, (b) 4area with s-FSC (Wang et al., 2023) (blue) and the proposed algorithm (orange). In each dataset, the left plot shows the runtime comparison and the right plot shows the average balance, for multiple numbers $k$ of clusters. Left plots show that our method is consistently faster than s-FSC, with even better efficiency gains as $k$ increases. Right plots compare the balance achieved by both methods, showing our method mantains the fairness in terms of balance of the exact algorithm.

Fair SC problem with $k = 25$ clusters. As shown in Table 2, our method consistently outperforms both o-FSC and s-FSC in terms of runtime across all datasets. For instance, on the LastFMNet dataset with $n = 5576$, our method shows a runtime of 4.59 seconds, which is significantly faster than both o-FSC (103.82 seconds) and s-FSC (19.08 seconds). Similar trends are observed for the other datasets, with our method achieving even higher speedups for larger datasets. For example, on the 4area dataset with $n = 35385$, our method takes 25.85 seconds compared to 166.92 seconds by s-FSC. Overall, o-FSC cannot complete the task within 500 seconds for the larger datasets (Census and 4area), and s-FSC takes considerably longer than our method.

Figure 3 compares runtime and average balance of the computed clustering by our method and s-FSC across multiple numbers of clusters $k \in [2, 50]$ on LastFMNet and 4area datasets. In terms of runtime, we observe that our method scales better in $k$ than s-FSC, especially in LastFMNet. This

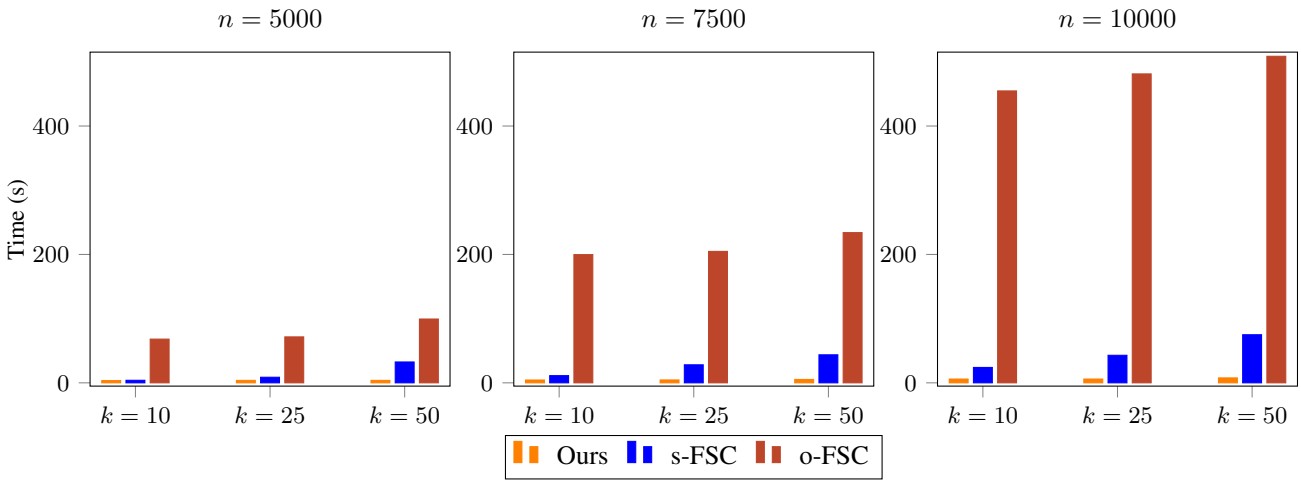

Figure 4: **Scalability**. Runtime (in seconds) of our algorithm, s-FSC, and o-FSC on RandLaplace at multiple sample sizes $n \in \{5000, 7500, 10000\}$ with $h = 5$ and $k \in \{10, 25, 50\}$.

Table 3: Comparison of clustering cost, balance, and time speedup vs. exact algorithm.

| Dataset | Method | Clustering | Balance | Time |
|---------|--------|-----------|---------|------|
| LastFM | s-FSC | $1.057_{\pm.063}$ | $0.0105_{\pm.0020}$ | $4.16\times$ |
| | Ours | $1.086_{\pm.074}$ | $0.0093_{\pm.0015}$ | $1\times$ |
| Thyroid | s-FSC | $0.353_{\pm.020}$ | $0.0029_{\pm.0005}$ | $4.13\times$ |
| | Ours | $0.353_{\pm.018}$ | $0.0030_{\pm.0004}$ | $1\times$ |
| Census | s-FSC | $134.539_{\pm3.667}$ | $0.0004_{\pm.0001}$ | $8.65\times$ |
| | Ours | $130.973_{\pm14.371}$ | $0.0004_{\pm.0001}$ | $1\times$ |
| 4area | s-FSC | $235.268_{\pm6.084}$ | $0.3884_{\pm.0150}$ | $6.45\times$ |
| | Ours | $242.000_{\pm10.303}$ | $0.3823_{\pm.0200}$ | $1\times$ |

Table 4: Comparison with methods FFSC and UFSC using different objectives for fair spectral clustering. Time is in seconds, "Cost" is the clustering cost, "Fairn." indicates the fairness constraint $\left\|F^\top H\right\|^2$, and "Ortho." indicates the orthogonality constraint $\left\|H^\top H - I\right\|^2$. Our method is significantly faster than both methods, while consistently achieving better fairness and orthogonality than FFSC.

| Dataset | Method | Time ($\downarrow$) | Cost ($\downarrow$) | Balance ($\uparrow$) | Fairn. ($\downarrow$) | Ortho. ($\downarrow$) |
|---------|--------|-----------|-----------|------------|------------|------------|
| LastFM | FFSC | 33.74 | 1067.91 | 0.2701 | 4.032 | 3.91E+00 |
| | UFSC | 16.36 | 3.54 | 0.016 | 1.21E-24 | 4.13E-13 |
| | Ours | 4.59 | 1.09 | 0.0093 | 0.000014 | 1.36E-11 |
| Thyroid | FFSC | 56.11 | 3080.85 | 0.0170 | 15.79 | 1.02E+01 |
| | UFSC | 95.08 | 0.94 | 0.0026 | 1.25E-24 | 3.77E-10 |
| | Ours | 7.38 | 0.35 | 0.0030 | 0.000001 | 2.18E-11 |
| Census | FFSC | 193.92 | 146669.89 | 0.0051 | 9.47 | 2.14E+00 |
| | UFSC | | | *timed out* | | |
| | Ours | 15.78 | 130.97 | 0.0004 | 0.000012 | 4.12E-10 |
| 4area | FFSC | 237.35 | 285174.39 | 0.6403 | 58.30 | 2.66E+00 |
| | UFSC | | | *timed out* | | |
| | Ours | 25.85 | 242.00 | 0.3823 | 0.000001 | 4.22E-10 |

trend shows that more iterations are needed for convergence of the additional eigenvectors in the eigendecomposition routines, while our method involves the computation of the SVD of a small $k \times k$ matrix; in the context of a small number of clusters, this SVD is computationally cheap. The plots for each dataset show that our method offers a more efficient solution for Fair SC on multiple real-world datasets while achieving comparable average balance than the competing exact algorithm.

We compare our method with the exact approach using eigendecomposition (s-FSC) in Table 3, where we report the clustering cost, a standard metric to evaluate spectral clustering (Shi & Malik, 2000), i.e., $\mathrm{Tr}(\hat{H}^\top M \hat{H})$ where $\hat{H}$ is computed from the 0-1 clustering indicator matrix obtained by applying $k$-means on $H$. The results show that our method consistently achieves comparable clustering costs and balance to the exact eigendecomposition approach across all datasets at a fraction of the runtime, with speedups

between $4 - 8\times$ faster than SOTA (Wang et al., 2023).

**Scalability in RandLaplace.** We use the RandLaplace dataset to illustrate the scalability of our method. Figure 4 presents the runtime of our method, s-FSC, and o-FSC for different sample sizes $n$ ranging from 5000 to 10000 with varying numbers of clusters $k \in \{10, 25, 50\}$. Our method requires less time than both compared methods across all clusters and sample sizes. These findings suggest that our method better scales with increasing sample size and cluster numbers, demonstrating the practicality of our method for real-world problems of varying sizes.

Table 5: Sensitivity analysis of penalty parameter $\alpha$.

| $\alpha$ | Iter. | Cost | Bal. | Fairn. Constr. | Ortho. Constr. |
|---|---|---|---|---|---|
| 0.005 | $9.0_{\pm.63}$ | $25.98_{\pm.004}$ | $1.0_{\pm.0}$ | $7.82\text{e-}08_{\pm5.49\text{e-}08}$ | $1.05\text{e-}14_{\pm6.55\text{e-}16}$ |
| 0.01 | $7.6_{\pm.49}$ | $25.99_{\pm.001}$ | $1.0_{\pm.0}$ | $8.86\text{e-}08_{\pm1.82\text{e-}08}$ | $1.07\text{e-}14_{\pm5.55\text{e-}16}$ |
| 0.05 | $6.6_{\pm.49}$ | $26.09_{\pm.001}$ | $1.0_{\pm.0}$ | $2.30\text{e-}07_{\pm7.82\text{e-}08}$ | $1.02\text{e-}14_{\pm6.73\text{e-}16}$ |
| 0.1 | $5.4_{\pm.49}$ | $26.06_{\pm.001}$ | $1.0_{\pm.0}$ | $1.27\text{e-}06_{\pm5.84\text{e-}07}$ | $9.92\text{e-}15_{\pm5.66\text{e-}16}$ |

**Comparative analysis with other spectral methods.** We compare with different formulations of spectral clustering with fairness constraints that deviate from (6) for comprehensive analysis. In particular, FFSC (Feng et al., 2024) and UFSC (Zhang & Wang, 2024) do not adopt the problem of (Kleindessner et al., 2019). FFSC instead introduces fairness by changing the SC objective with a different regularization term. UFSC learns an induced fairer graph where eigendecomposition is applied. Comparative results are shown in Table 4. Our method is $7-12\times$ faster and achieves significantly better spectral clustering cost. FFSC's modified objective shows a different balance-clustering trade-off with higher balance but worse clustering and group fairness. Note the two different measures of fairness, where balance promotes same number of individuals in each cluster (3), and group fairness ensures proportional representation (4). Overall, these methods' solutions are far from the exact one of (6) leading to higher spectral cost, potential instability when orthogonality is not enforced, and longer runtimes.

**Sensitivity analysis of $\alpha$.** We conduct a sensitivity analysis of our algorithm w.r.t. the ADMM penalty parameter $\alpha$. Table 5 presents key indicators obtained on the RandLaplace dataset with $n = 1000, k = 25$ over 5 runs, detailing how varying the initial $\alpha$ influences the number of ADMM iterations, the final objective cost, the attained balance, and the final feasibility of the orthogonality and linear fairness constraints. All tested $\alpha$ values lead to solutions with similar final cost and constraint satisfaction, as well as perfect balance (1.0) for the studied dataset. This observation suggests that our algorithm is robust to changes in $\alpha$. As $\alpha$ increases, the faster convergence due to higher penalty on primal feasibility comes at the cost of slightly worse fairness constraint satisfaction. Given this slight decrease with larger $\alpha$, in our experiments we opt for a smaller $\alpha^{(0)} = 0.005$ and update it as detailed in Appendix B.

## 5. Conclusion

We design a significantly faster algorithm for Fair SC than previous state-of-the-art. Our main contributions lie in $(i)$ reformulating the Fair SC problem into a differences of convex functions problem where we avoid expensive matrix routines using $M^2$ instead of $M^{1/2}$, and $(ii)$ showing that the ADMM subproblems can be solved efficiently, thanks to the DC dualization enabled by our design choice of includ-

ing the fairness constraints as $MH$ instead of $H$. Empirical evaluations show significant speedups with comparable clustering and fairness metrics to exact algorithms. Future works include investigating model-based formulations within kernel-based settings and applying our framework to more general constrained spectral clustering by accommodating other constraints whenever the DC formulation is maintained. For example, it might be possible to extend our framework to constrained spectral clustering, e.g., linear constraints for must-link and cannot-link constraints, by considering a modified indicator function $h(\cdot)$, accomodating other constraints or fairness metrics as DC functions. ADMM could then be applied with appropriate modifications to the subproblem w.r.t. $Y$.

## Impact Statement

A faster fair spectral clustering algorithm can have a significantly positive societal impact as it can facilitate a more widespread adoption of fairer clusterings in applications with larger real-word datasets. The approach presented in this paper aims at advancing the field of Machine Learning. No other potential societal consequences of our work are deemed necessary to specifically highlight here.

## Acknowledgements

LIONS-EPFL was supported by Hasler Foundation Program: Hasler Responsible AI (project number 21043). LIONS-EPFL was sponsored by the ARO under Grant Number W911NF-24-1-0048. LIONS-EPFL was supported by the Swiss National Science Foundation (SNSF) under grant number 200021_205011. ESAT-STADIUS was suported by the Flemish Government (AI Research Program); iBOF/23/064; KU Leuven C1 project C14/24/103.

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

# A. Additional experimental results

## A.1. Real-world data

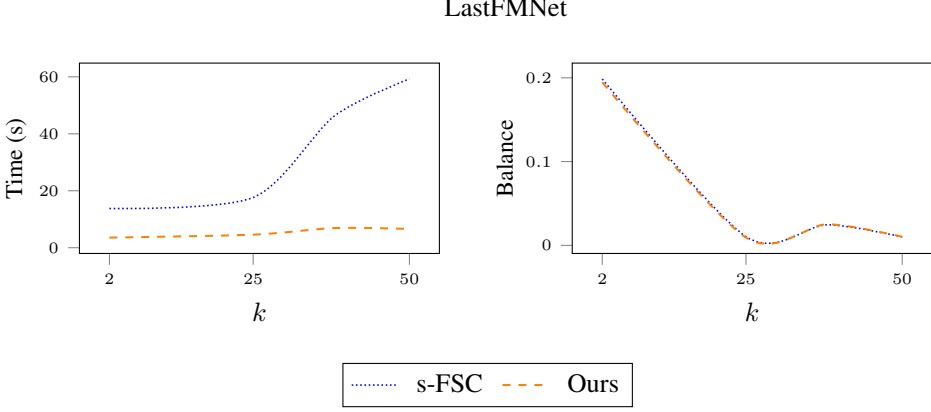

Figure 5: Fair spectral clustering on LastFMNet with s-FSC (Wang et al., 2023) (blue) and the proposed algorithm (orange). The plot uses the same structure as Figure 3 in the main body.

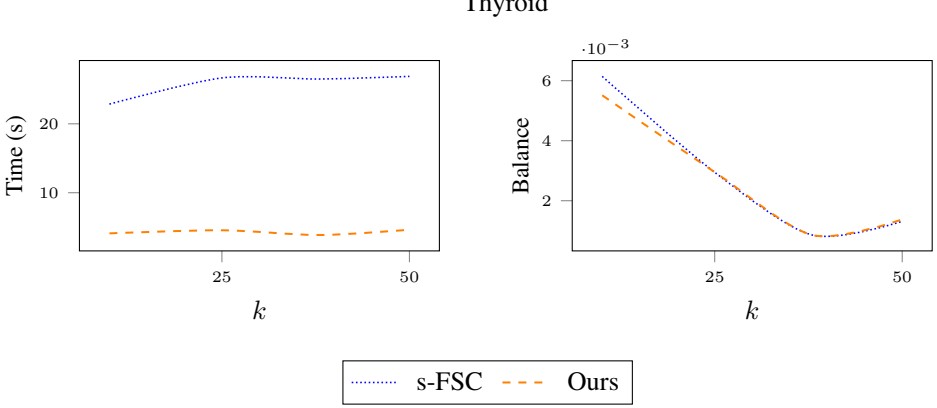

Figure 6: Fair spectral clustering on Thyroid with s-FSC (Wang et al., 2023) (blue) and the proposed algorithm (orange). The plot uses the same structure as Figure 3 in the main body.

In this experiment, we evaluate the runtime and average balance of the proposed algorithm for Fair SC on the real-world datasets summarized in Table 10. The affinity graphs for Thyroid, Census, and 4area are obtained by a radial basis function (RBF) kernel $k(x_i, x_j) = \exp(-\gamma \|x_i - x_j\|^2)$, where $\gamma = 1/d$ and $\gamma = 1/d * 0.01$ for the smaller Thyroid, with $x_i \in \mathbb{R}^d$, $i = 1, \ldots, n$. We compare with the fastest available Fair SC algorithm, i.e., s-FSC from (Wang et al., 2023). Here, we report the complete results for the LastFMNet, Thyroid, 4area, and Census datasets in Figures 5 to 8.

We compare the runtime and average balance of the computed clustering across multiple numbers of clusters $k \in \{2, \ldots, 50\}$. Our method consistently outperforms state-of-the-art s-FSC in terms of runtime across all datasets while maintaining approximately the same level of balance. Our method overall scales better in $k$ than s-FSC, especially in LastFMNet. Both methods require a longer time for larger $k$ values. s-FSC requires more iterations for convergence of the additional eigenvectors in the eigendecomposition routines, while our method involves the computation of the SVD of a small $k \times k$ matrix; this is computationally cheap for a small number of clusters, which is usually the case in applications. Our method also involves the $MH$ matrix product with complexity $\mathcal{O}(n^2 k)$ for dense $M$; however, matrix multiplication is in practice significantly more efficient than the eigendecomposition. In terms of average balance, the clustering computed by our method achieves approximately the same balance to s-FSC. Balance on Thyroid and Census is low for both algorithms, which means that, in the Fair SC clustering, some groups are under-represented across multiple clusters. Overall, our method

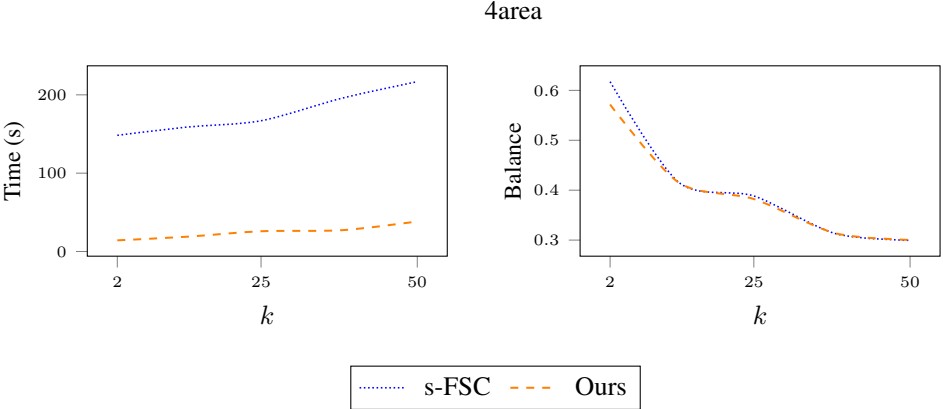

Figure 7: Fair spectral clustering on 4area with s-FSC (Wang et al., 2023) (blue) and the proposed algorithm (orange). The plot uses the same structure as Figure 3 in the main body.

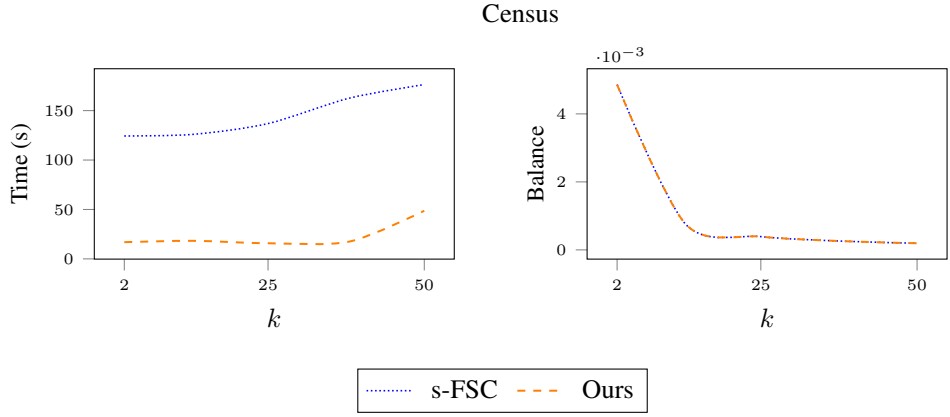

Figure 8: Fair spectral clustering on Census with s-FSC (Wang et al., 2023) (green) and the proposed algorithm (blue). The plot uses the same structure as Figure 3 in the main body.

offers a more efficient solution for Fair SC on real-world datasets, without compromising the fairness of the results w.r.t. the exact Fair SC solution. This makes it a promising choice for applications where both efficiency and fairness are crucial.

### A.2. Scalability in RandLaplace

In this experiment, we employ the RandLaplace dataset to evaluate the scalability of our proposed algorithm w.r.t. sample size $n$ at different $k$. The RandLaplace dataset is generated by a random $n \times n$ symmetric adjacency matrix and $h = 2$ protected groups randomly assigned according to a Bernoulli distribution with probability 0.3. The performance comparison between our approach and the existing Fair SC methods, o-FSC (Kleindessner et al., 2019) and s-FSC (Wang et al., 2023), is depicted in Figure 4 and reported for completeness in Table 6 here. The results show that our algorithm outperforms the compared ones in terms of computational time for all tested sample sizes and cluster sizes. These results indicate that our algorithm exhibits superior scalability as the data size and number of clusters increase, underscoring our enhanced applicability for larger-scale problems.

### A.3. Additional visualizations on synthetic data

We provide additional figures of the clustering results to illustrate the effectiveness of our method to preserve the clustering structure while satisfying the fairness constraints. We consider the 2D datasets from (Feng et al., 2024): the Elliptical dataset with $k = 2, h = 2$ and the DS-577 dataset with $k = 3, h = 3$. Figure 9 shows the clustering results on the Elliptical

Table 6: Running time in seconds for the RandLaplace dataset with different configurations (number of samples $n \in \{5000, 7500, 10000\}$ and clusters $k \in \{10, 25, 50\}$). Standard deviations in parentheses.

(a) $k = 10$, $h = 5$

| $n$ | o-FSC | s-FSC | Ours |
|---|---|---|---|
| 5000 | 68.12 (2.48) | 4.00 (0.09) | **3.50** (0.10) |
| 7500 | 199.72 (3.20) | 11.22 (0.27) | **4.42** (0.12) |
| 10000 | 454.68 (5.50) | 23.99 (0.31) | **5.93** (0.23) |

(b) $k = 25$, $h = 5$

| $n$ | o-FSC | s-FSC | Ours |
|---|---|---|---|
| 5000 | 71.58 (2.0) | 8.73 (0.30) | **3.84** (0.09) |
| 7500 | 204.72 (4.62) | 28.10 (2.25) | **4.52** (0.16) |
| 10000 | 481.18 (6.91) | 42.93 (3.91) | **5.93** (0.10) |

(c) $k = 50$, $h = 5$

| $n$ | o-FSC | s-FSC | Ours |
|---|---|---|---|
| 5000 | 99.41 (3.0) | 32.52 (2.11) | **3.85** (0.09) |
| 7500 | 233.97 (8.35) | 43.66 (7.36) | **5.29** (0.11) |
| 10000 | 508.63 (9.78) | 74.94 (7.62) | **7.70** (0.16) |

Table 7: Running time in seconds (average and standard deviation over 5 runs) for the m-SBM benchmark from Table 1 and the considered real-world datasets from Table 2.

| Dataset | Time (s) | | |
|---|---|---|---|
| | o-FSC | s-FSC | Ours |
| m-SMB ($n = 5000$) | 96.39 (3.62) | 75.61 (2.58) | **3.29** (0.11) |
| m-SMB ($n = 7500$) | 235.62 (4.48) | 88.99 (4.35) | **5.29** (0.21) |
| m-SMB ($n = 10000$) | 495.70 (9.49) | 107.94 (5.10) | **9.19** (0.36) |
| LastFMNet | 103.82 (2.87) | 19.08 (1.96) | **4.59** (0.30) |
| Thyroid | 279.03 (4.21) | 30.49 (3.17) | **7.38** (0.16) |
| Census | - | 136.60 (0.69) | **15.78** (1.06) |
| 4area | - | 166.92 (0.73) | **25.85** (1.4) |

dataset (top) and the DS-577 dataset (bottom). These plots show that our method produces assignments comparable to exact algorithms (o-FSC,s-FSC). Critically, we achieve this with reduced computations, as shown in the main paper, which constitutes our main contribution.

## A.4. Additional metrics and dataasets

In main body, we follow the setups of (Kleindessner et al., 2019; Wang et al., 2023) and report the average balance. For completeness, we also report the minimum balance for Table 2 with $k = 25$ in Table 8.

Table 8: Minimum balance for the considered real-world datasets from Table 2.

| Dataset | Time (s-FSC) | Time (Ours) | Min. Balance (s-FSC) | Min. Balance (Ours) |
|---|---|---|---|---|
| LastFM | 19.08 | 4.59 | 0.0027 | 0.0029 |
| Thyroid | 30.49 | 7.38 | 0.0011 | 0.0012 |
| Census | 136.60 | 15.78 | 0.0001 | 0.0001 |
| 4area | 166.92 | 25.85 | 0.1582 | 0.1517 |

We also evaluate on the common FacebookNet dataset (Wang et al., 2023) that collects Facebook friendship links with $n = 155, h = 2$. We show results in Table 9. These results further demonstrates the computational advantage of our method, while achieving similar clustering quality to exact algorithms.

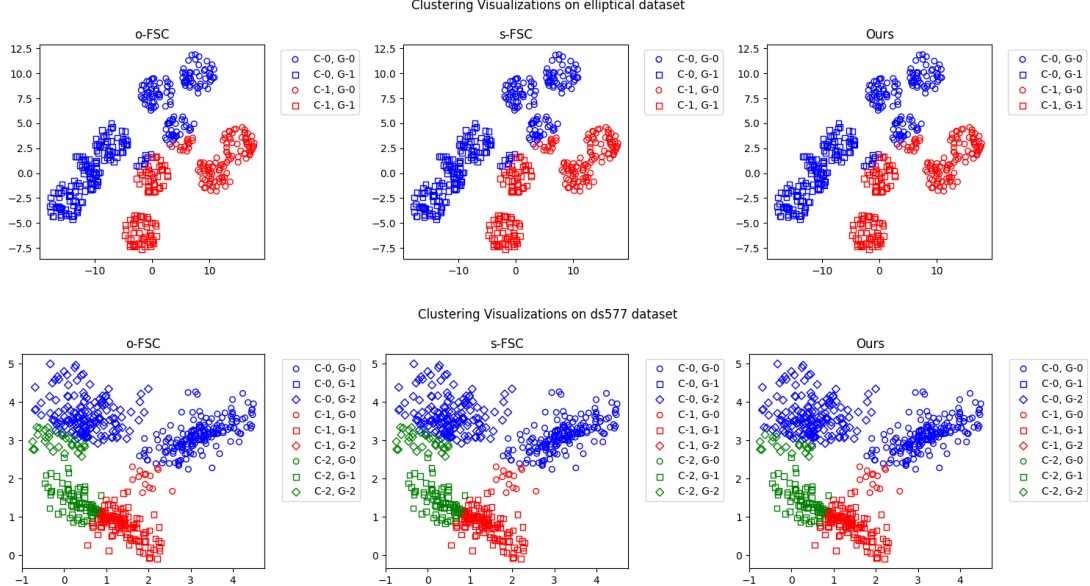

Figure 9: Clustering results on synthetic datasets. The top panel shows the Elliptical dataset with $k = 2, h = 2$, and the bottom panel shows the DS-577 dataset with $k = 3, h = 3$. Our method preserves the clustering structure while satisfying fairness constraints.

Table 9: Running time in seconds for the FacebookNet dataset with balance and spectral clustering cost.

| #Clusters | Method | Time (s) | Balance | SC Cost |
|-----------|--------|----------|---------|---------|
| 2 | o-FSC | 0.798 | 1.00 | 0.126 |
| 2 | s-FSC | 0.193 | 1.00 | 0.126 |
| 2 | Ours | 0.055 | 1.00 | 0.133 |
| 25 | o-FSC | 9.908 | 0.84 | 14.114 |
| 25 | s-FSC | 6.422 | 0.84 | 14.114 |
| 25 | Ours | 0.131 | 0.84 | 14.128 |
| 50 | o-FSC | 12.243 | 0.58 | 37.084 |
| 50 | s-FSC | 11.558 | 0.58 | 37.084 |
| 50 | Ours | 0.215 | 0.58 | 37.100 |

## B. Additional experimental details

**Experimental setups.**   Regarding optimization of the dual DC problem, we employ the LBFGS algorithm in the scipy implementation with backtracking line search using the strong Wolfe conditions with initialization from the standard normal distribution; the stopping condition is given by `gtol` $= 10^{-3}$, `ftol` $= 10^{-4}$. The ADMM algorithm is run with $\alpha^{(0)} = 0.005, T = 10$. The compared methods o-FSC (Kleindessner et al., 2019) and s-FSC (Wang et al., 2023) employ the scipy eigendecomposition routine. For s-FSC, we use $\left\| \hat{L} \right\|_1$ as shift, as suggested in their paper. The $k$-means algorithm is run to obtain the clustering indicators for all compared methods. The real-world datasets used in the experiments are summmarized in Table 10.

The update rule for $\alpha$ in Algorithm 1 is standard in ADMM (Boyd et al., 2011) and is given by

$$\alpha^{(i+1)} = \begin{cases} \tau\alpha^{(i)} & \text{if } \left\|R^{(i)}\right\|_F > \mu\left\|S^{(i)}\right\|_F \\ \alpha^{(i)}/\tau & \text{if } \left\|S^{(i)}\right\|_F > \mu\left\|R^{(i)}\right\|_F \;, \\ \alpha^{(i)} & \text{otherwise} \end{cases} \tag{19}$$

with primal and dual residuals $R^{(i)} = MH^{(i)} - Y^{(i)}$ and $S^{(i)} = \alpha^{(i)}(Y^{(i)} - Y^{(i+1)})$. The parameters $\tau$ and $\mu$ are set to 2

Table 10: Description of the real-world datasets used for the experiments.

| Dataset | Samples ($n$) | Groups ($h$) | Group type |
|---------|---------------|--------------|------------|
| LastFMNet | 5576 | 6 | Nationality |
| Thyroid | 7200 | 3 | Disease |
| Census | 32561 | 7 | Demographic |
| 4area | 35385 | 4 | Research field |

and 10, respectively.

## C. Additional discussions on related works

Our method follows the established line of work on Fair SC, which enforces fairness via linear constraints in the embedding (Kleindessner et al., 2019; Wang et al., 2023). There exist other spectral clustering methods that do not follow this line of work and enforce fairness via different means. (Feng et al., 2024) introduce fairness by changing the SC objective with a different fairness regularization term; they do not employ the group-fairness constraint $F^\top H = 0$. They apply coordinate descent to this new objective where they relax the discrete clustering indicator constraint without orthogonality constraints $H^\top H = I$. Therefore, (Feng et al., 2024) solves a different problem than (6): their solution is far from the exact one leading to higher spectral cost and potential training instability. Another related work is (Zhang & Wang, 2024). Our work designs a much faster method for the existing Fair SC problem defined in (Kleindessner et al., 2019). (Zhang & Wang, 2024) instead focuses on improving fairness and is orthogonal to our algorithmic contribution: it learns an induced fairer graph where Fair SC is applied. Future work could combine our method on top of (Zhang & Wang, 2024) by applying our faster algorithm to their learned graph.

Other works on fair clustering include (Bera et al., 2019b; Backurs et al., 2019), which are not directly comparable to our work as they do not consider the spectral clustering objective. (Bera et al., 2019b) propose LP-based $k$-(means, median, center) clustering. Using their formulation in our work would ignore the RatioCut objective, resulting in suboptimal solutions wr.t. the spectral objective. (Backurs et al., 2019) use fairlets for prototype-based clustering. However, extending the fairlet analysis, which relies on the $k$-median and $k$-center cost of the fairlet decomposition, to the spectral setting is not trivial. SC involves a spectral embedding step followed by a clustering in the embedding space, where reassigning points within a fairlet can significantly alter the spectral embedding and hence potentially violating fairness, making it non-trivial to incorporate their analysis in the fair SC case. The primary contribution of our present work is to design a significantly faster method for the already established Fair SC problem defined in (Kleindessner et al., 2019), rather than designing alternative fair clustering problems.

## D. Proofs and derivations

### D.1. Proofs of Assumptions of Proposition 3.4

We prove the convergence result from Proposition 3.4 by applying Proposition 3 from Magnusson et al. (2016) to our specific ADMM structure. There they consider problems of the form

$$\min_{x,z} \quad u(x) + v(z) \tag{20}$$

$$\text{s.t.} \quad x \in \mathcal{X}, z \in \mathcal{Z} \tag{21}$$

$$Ax + Bz = c. \tag{22}$$

We can already cast Problem 11 into this structure by letting

- $\mathcal{X} = \mathcal{S}_n^k$ and $u(x) = -f(Mx)$

- $\mathcal{Z} = \{z \in \mathbb{R}^{n \times k} \mid F^\top Y = 0\}$ and $v(z) = 0$

- $A = M$, $B = -I$ and $c = 0$.

To apply their result, we verify that four key assumptions hold, namely Assumption D.1, Assumption D.2, Assumption D.4 and Assumption D.5 that can be found in Magnusson et al. (2016).

**Assumption D.1.** The functions $u$ and $v$ are continuously differentiable.

This is satisfied in our setting.

**Assumption D.2.** The sets $\mathcal{X}$ and $\mathcal{Z}$ are closed and can be expressed in terms of a finite number of equality and inequality constraints. In particular,

$$\mathcal{X} = \{x \in \mathbb{R}^{n \times k} \mid \psi(x) = 0, \ \phi(x) \leq 0\} \tag{23}$$

$$\mathcal{Z} = \{z \in \mathbb{R}^{n \times k} \mid \theta(z) = 0, \ \sigma(z) \leq 0\} \tag{24}$$

where $\psi, \phi, \theta, \sigma$ are continuously differentiable functions.

This is clear from the definition of the spaces $\mathcal{X}$ and $\mathcal{Z}$, a proof is provided for the case of the Stiefel manifold in the following lemma.

**Lemma D.3** (Smooth Representation of Orthogonal Matrix Constraints). *. Let $A \in \mathbb{R}^{n \times m}$ be a matrix satisfying the orthogonality constraint $A^\top A = I_m$, where $I_m$ denotes the $m \times m$ identity matrix. Then there exists a smooth function $\psi : \mathbb{R}^{nm} \to \mathbb{R}^{m(m+1)/2}$ such that the constraint can be equivalently expressed as*

$$\{x \in \mathbb{R}^{nm} : \psi(x) = 0\},$$

*where $x = vec(A)$ is the vectorization of matrix $A$.*

*Proof.* Let $x \in \mathbb{R}^{nm}$ denote the vectorization of matrix $A$ obtained by column-wise stacking:

$$x = \begin{bmatrix} a_{11} \\ \vdots \\ a_{n1} \\ a_{12} \\ \vdots \\ a_{nm} \end{bmatrix}.$$

Under this vectorization scheme, the $(k, j)$-th element of $A$ corresponds to the $((j-1)n + k)$-th component of $x$, i.e., $a_{kj} = x_{(j-1)n+k}$. The orthogonality constraint $A^\top A = I_m$ is equivalent to requiring that the columns of $A$ form an orthonormal system. Specifically, for all $1 \leq i \leq j \leq m$:

$$\sum_{k=1}^{n} a_{ki} a_{kj} = \delta_{ij},$$

where $\delta_{ij}$ denotes the Kronecker delta. Expressing this constraint in terms of the vectorized representation $x$, we obtain:

$$\sum_{k=1}^{n} x_{(i-1)n+k} x_{(j-1)n+k} = \delta_{ij}.$$

We now define the function $\psi : \mathbb{R}^{nm} \to \mathbb{R}^{m(m+1)/2}$ with components:

$$\psi_{ij}(x) = \sum_{k=1}^{n} x_{(i-1)n+k} x_{(j-1)n+k} - \delta_{ij}, \quad 1 \leq i \leq j \leq m.$$

The orthogonality constraint $A^\top A = I_m$ is then equivalent to the condition $\psi(x) = 0$. To establish smoothness, observe that each component $\psi_{ij}(x)$ is a quadratic polynomial in the elements of $x$. Since polynomial functions possess derivatives of all orders, each $\psi_{ij}$ is smooth (i.e., $C^\infty$). Consequently, the vector-valued function $\psi$ is smooth. Therefore, the orthogonality constraint on matrix $A$ admits a smooth representation. $\square$

**Assumption D.4.** At every step from Algorithm 1, the solutions of sub-Problem 13 and sub-Problem 14 computed are locally or globally optimal.

This assumption allows solutions to the subproblems to be only locally optimal. In our case, both subproblems allow global optimality.

**Assumption D.5.** Let $\mathcal{L}$ denote the set of limit points of the sequence $\{x_i := H^{(i)}, z_i = Y^{(i)}\}_{i \in \mathbb{N}}$ and let $(\bar{x}, \bar{z}) \in \mathcal{L}$. The set of constraint gradient vectors at $\bar{x}$,

$$\mathcal{C}_{\mathcal{X}}(\bar{x}) = \{\nabla \psi_i(\bar{x}) | i = 1, \dots, \dim \psi\} \cup \{\nabla \phi_i(\bar{x}) | i \text{ s.t. } \phi_i(\bar{x}) = 0\} \tag{25}$$

associated to $\mathcal{X}$ is linearly independent. Similarly, the set of constraint gradient vectors $\mathcal{C}_{\mathcal{Z}}(\bar{z})$ is linearly independent.

This assumption is a regularity assumption that is usually satisfied in practice according to Magnusson et al. (2016). The following lemmas are dedicated to proving that it holds in our case.

**Lemma D.6** (Linear Independence of Orthogonal Constraint Gradients). *Let $x \in \mathbb{R}^{nm}$ be identified with the matrix $A \in \mathbb{R}^{n \times m}$, where $A_{pi} = x_{pi}$ for $p = 1, \dots, n$ and $i = 1, \dots, m$. Define $\psi_{ij}(x) = \sum_{p=1}^{n} x_{pi} x_{pj} - \delta_{ij}$ for $1 \le i \le j \le m$. Let $C(x) = \{\nabla_x \psi_{ij}(x) | 1 \le i \le j \le m\}$. If $A^T A = I$, then $C(\bar{x})$ is linearly independent at $\bar{x}$.*

*Proof.* Suppose there exist scalars $\lambda_{ij}$ (with $\lambda_{ij} = \lambda_{ji}$) such that

$$\sum_{1 \le i \le j \le m} \lambda_{ij} \nabla_x \psi_{ij}(\bar{x}) = 0.$$

Let $\Lambda$ be the $m \times m$ symmetric matrix with entries $\Lambda_{ij} = \lambda_{ij}$. Define the scalar function

$$f(A) = \sum_{1 \le i \le j \le m} \lambda_{ij} \psi_{ij}(x) = \sum_{i,j=1}^{m} \frac{\lambda_{ij}}{2} (a_i^T a_j - \delta_{ij}) = \frac{1}{2} \mathrm{Tr}(\Lambda(A^T A - I)),$$

where $a_i$ denotes the $i$-th column of $A$. For any perturbation $H \in \mathbb{R}^{n \times m}$, the directional derivative of $f$ at $A$ in the direction of $H$ is

$$\begin{aligned} Df(A)[H] &= \frac{1}{2} \mathrm{Tr}(\Lambda(H^T A + A^T H)) \\ &= \mathrm{Tr}(H^T A \Lambda) \\ &= \langle \mathrm{vec}(H), \mathrm{vec}(A\Lambda) \rangle. \end{aligned}$$

By the chain rule, and since we assumed $\sum \lambda_{ij} \nabla_x \psi_{ij}(\bar{x}) = 0$, we have

$$Df(A)[H] = \left\langle \sum_{1 \le i \le j \le m} \lambda_{ij} \nabla_x \psi_{ij}(A), \mathrm{vec}(H) \right\rangle = 0, \quad \forall H.$$

This implies $\mathrm{vec}(A\Lambda) = 0$, and hence $A\Lambda = 0$. Since $A^T A = I$ at $\bar{x}$, $A$ has full column rank, and thus $\Lambda = 0$. This implies $\lambda_{ij} = 0$ for all $i, j$. Therefore, the only linear combination of the gradients $\nabla_x \psi_{ij}(\bar{x})$ that equals zero is the trivial one, which proves that $C(\bar{x})$ is linearly independent. $\square$

**Lemma D.7.** *Let $F \in \mathbb{R}^{n \times h}$ and $H \in \mathbb{R}^{n \times k}$. Define $z = \mathrm{vec}(H) \in \mathbb{R}^{nk}$ and consider the constraint $F^\top H = 0$. Let $\theta(z) = \mathrm{vec}(F^\top H) \in \mathbb{R}^{hk}$. The set of constraint gradients*

$$C(z) = \{\nabla_z \theta_i(z) \mid i = 1, \dots, hk\}$$

*is linearly independent at any $\bar{z}$ such that $F^\top H = 0$ whenever $F$ has full column rank.*

*Proof.* Using the Kronecker product identity $\mathrm{vec}(AB) = (I \otimes A) \mathrm{vec}(B)$, we can express $\theta(z)$ as:

$$\theta(z) = (I_k \otimes F^\top)z = Mz,$$

where $M = I_k \otimes F^\top \in \mathbb{R}^{hk \times nk}$.

Since $\theta(z)$ is linear, its Jacobian is the constant matrix $M$. The gradient of the $i$-th component of $\theta(z)$, denoted $\nabla_z \theta_i(z)$, is the $i$-th row of $M$ transposed. Thus, the set $C(z)$ consists of the rows of $M$.

The rows of $M$ are linearly independent if and only if $M$ has full row rank, which is $hk$. We have:

$$\operatorname{rank}(M) = \operatorname{rank}(I_k) \operatorname{rank}(F^\top) = k \operatorname{rank}(F).$$

Therefore, $\operatorname{rank}(M) = hk$ if and only if $\operatorname{rank}(F) = h$. This means the gradients in $C(z)$ are linearly independent if and only if $F$ has full column rank (if $\operatorname{rank}(F) = h$). For instance, this is the case in the Fair SC problem when $h$ is set to be the number of groups minus 1 (Wang et al., 2023). This can be guaranteed by solving (6) with $F := F[:, :h-1]$, i.e., removing the last column of $F$, which guarantees that $\operatorname{rank}(F) = h-1$ with same range (Wang et al., 2023). This holds regardless of the order of the groups (Wang et al., 2023). $\qquad\square$

### D.2. Derivation of Problem 6

In this subsection, we review the derivation of the fair spectral clustering problem (6). First, recall that spectral clustering aims at partitioning $\mathcal{D}$ into $k$ clusters by minimization of a normalized cut objective, as detailed in (Shi & Malik, 2000; Von Luxburg, 2007). The spectral clustering problem is defined as

$$\min_{T \in \mathbb{R}^{n \times k}} \operatorname{Tr}(T^\top L T) \quad \text{s.t.} \quad T^\top D T = I_k. \tag{26}$$

Because of the normalized cut objective, Problem 26 is also often referred to as the *normalized* spectral clustering problem, to distinguish it from the spectral clustering problem with mincut criterion without normalization. By substituting $T = D^{-1/2} H$ in Problem 26, we obtain the equivalent problem in terms of the normalized Laplacian $\hat{L}$:

$$\min_{H \in \mathbb{R}^{n \times k}} \operatorname{Tr}(H^\top \hat{L} H) \quad \text{s.t.} \quad H^\top H = I_k. \tag{27}$$

Solving the spectral clustering problem amounts to finding the $k$ eigenvectors of $\hat{L}$ corresponding to the $k$ smallest eigenvalues and then applying $k$-means clustering to the rows of $D^{-1/2} H$. The fair spectral clustering problem is obtained by adding the fairness constraint (5) to the spectral clustering problem (26):

$$\min_{T \in \mathbb{R}^{n \times k}} \operatorname{Tr}(T^\top L T) \quad \text{s.t.} \quad T^\top D T = I_k, \quad F_0^\top T = 0. \tag{28}$$

By substituting $T = D^{-1/2} H$ as above, we obtain the equivalent Fair SC problem in terms of the normalized Laplacian $\hat{L}$:

$$\min_{H \in \mathbb{R}^{n \times k}} \operatorname{Tr}(H^\top \hat{L} H) \quad \text{s.t.} \quad H^\top H = I_k, \quad F^\top H = 0, \tag{29}$$

where $F = D^{-1/2} F_0$. Now, we have that for all $H \in \mathbb{R}^{n \times k}$,

$$\operatorname{Tr}(H^\top \hat{L} H) = \operatorname{Tr}\left(H^\top H\right) - \operatorname{Tr}\left(H^\top D^{-1/2} W D^{-1/2} H\right) = \operatorname{Tr}\left(H^\top H\right) - \operatorname{Tr}\left(H^\top M H\right).$$

On the Stiefel manifold, the quantity $\operatorname{Tr}\left(H^\top H\right)$ is constant so that the minimizers of Problem 29 are the maximizers of Problem 6.

### D.3. Proof of Proposition 3.2

*Proof.* Expanding $\phi(MH)$ to recover the expression of the augmented Lagrangian shows that $H \mapsto \phi(MH)$ is convex as long as $\alpha < 1$. Then, following Proposition 3.1 from (Tonin et al., 2023), equivalently we can write the problem as

$$\inf_{H \in \mathbb{R}^{n \times k}} \left[ g(H) - \sup_{V \in \mathbb{R}^{n \times k}} \{\langle V, MH \rangle - \phi^*(V)\} \right] = \inf_{H \in \mathbb{R}^{n \times k}} g(H) + \inf_{V \in \mathbb{R}^{n \times k}} \{\phi^*(V) - \langle V, MH \rangle\} \tag{30}$$

$$= \inf_{H \in \mathbb{R}^{n \times k}, V \in \mathbb{R}^{n \times k}} g(H) + \phi^*(V) - \langle V, MH \rangle \tag{31}$$

$$= \inf_{V \in \mathbb{R}^{n \times k}} \phi^*(V) - \sup_{H \in \mathbb{R}^{n \times k}} \{\langle MV, H \rangle - g(H)\} \tag{32}$$

$$= \inf_{V \in \mathbb{R}^{n \times k}} \phi^*(V) - g^*(MV), \tag{33}$$

where we used the self-adjointness of $M$.

$\square$

### D.4. Proof of Proposition 3.3

*Proof.* We begin by the derivation of $\phi^\star(V)$:

$$\phi^\star(V) = \sup_Z \{\langle V, Z \rangle - \phi(Z)\} \tag{34}$$

$$= \sup_Z \{\langle V, Z \rangle - \frac{1}{2} \|Z\|^2 + \langle P, Z \rangle + \frac{\alpha}{2} \|Z - Y\|^2\} \tag{35}$$

$$= \sup_Z \{-\frac{1}{2} \|Z - V\|^2 + \frac{1}{2} \|V\|^2 + \langle P, Z \rangle + \frac{\alpha}{2} \|Z - Y\|^2\} \tag{36}$$

$$= \frac{1}{2} \|V\|^2 - \inf_Z \{\frac{1}{2} \|Z - V\|^2 - \langle P, Z \rangle - \frac{\alpha}{2} \|Z - Y\|^2\}. \tag{37}$$

By nullifying the gradient in $Z$, we find that the critical point $\hat{Z}$ satisfies $\hat{Z} = \eta(V + P - \alpha Y)$, where $\eta = \frac{1}{1-\alpha}$, which in turn gives

$$\phi^\star(V) = \frac{1}{2} \|V\|^2 - \frac{1}{2} \|\eta(V + P - \alpha Y) - V\|^2 \tag{38}$$

$$+ \langle P, \eta(V + P - \alpha Y) \rangle \tag{39}$$

$$+ \frac{\alpha}{2} \|\eta(V + P - \alpha Y) - Y\|^2. \tag{40}$$

We now prove that $g^\star(MV) = \mathrm{Tr}\left(\sqrt{V^\top M^2 V}\right)$ by using that the Fenchel-Legendre conjugate of $g$ is the Schatten 1-norm, also called the nuclear norm: $g^\star(U) = \sum_{i=1}^k |\lambda(U)_i| := \|U\|_{S_1}$. We then have

$$\sup_{H^\top H \preceq I_k} \langle MV, H \rangle = \|MV\|_{S_1} = \mathrm{Tr}\left(\sqrt{V^\top M^2 V}\right), \tag{41}$$

where the last equality is obtained by exploiting the SVD of $MV$. $\square$

