# OpenReview forum: "Accelerating Spectral Clustering under Fairness Constraints"
_ICML.cc/2025/Conference — ICML 2025 poster_

### Official Review · Reviewer_ztMM · 2025-03-04

**Overall Recommendation:** 3

**Summary:**

- This paper proposes a computationally efficient algorithm for the fair spectral clustering problem.
- The key of the proposed algorithm is the use of DC (Difference of Convex functions), which leads to the ADMM framework.
- The authors claim that the proposed method is empirically faster than two existing algorithms, o-FSC and s-FSC.
- The main distinction is that the proposed algorithm is based on gradient-based optimization, whereas the two existing algorithms rely on eigendecomposition routines.

## update after rebuttal
- Thank you for the clarifications. I will maintain my rating.

**Claims And Evidence:**

- Overall, the claims are supported by the experiments.
- The gap between the theoretical and empirical complexities of s-FSC and the proposed algorithm could be discussed more thoroughly (see **Questions For Authors** below for details).

**Essential References Not Discussed:**

- Fairness in constrained spectral clustering (https://doi.org/10.1016/j.neucom.2025.129815)
   - This work was very recently accepted. However, I believe the authors should at least conceptually compare their approach with this study.

**Ethical Review Concerns:**

- N/A

**Experimental Designs Or Analyses:**

- Questions about the comparison between s-FSC and the proposed method:
   - Figure 2: For $k = 2$, neither s-FSC nor the proposed method appears to achieve high balance (0.2 for LastFMNet and over 0.5 for the 4area dataset), which is far from the perfect fairness (balance = 1). Can these results be considered fair?
- Additional experiments/metrics would enhance the practical effectiveness of the proposed algorithm:
   - Baselines: While the objective of the proposed algorithm aligns closely with those of o-FSC and s-FSC, given that the paper considers the spectral clustering for group fairness, it would be beneficial to compare it with other existing algorithms for group fairness. Examples are:
      - Bera et al. (2019) https://proceedings.neurips.cc/paper_files/paper/2019/file/fc192b0c0d270dbf41870a63a8c76c2f-Paper.pdf,
      - Backurs et al. (2019) https://proceedings.mlr.press/v97/backurs19a/backurs19a.pdf
   - Metrics: - The average balance metric used is a relaxed version of the commonly adopted balance metric in the fair clustering literature, which is defined as the minimum of Eq. (3) over the cluster indices $l \in [k]$. Computing the minimum balance would strengthen the experiment section of this paper.

**Methods And Evaluation Criteria:**

- Several experimental setups (e.g., metrics) are aligned with those used in existing methods.
- The FacebookNet dataset was used in both Kleindessner et al. (2019) and Wang et al. (2023); however, this paper does not consider it.

**Other Comments Or Suggestions:**

- N/A

**Other Strengths And Weaknesses:**

- N/A

**Questions For Authors:**

- Complexity of s-SFC and the proposed algorithm:
   - While Sections 2.2.2 and 3.2 claim that both s-FSC and the proposed algorithm theoretically have a complexity of $O(n^2)$, the experimental results indicate that the proposed algorithm significantly reduces computation time on real-world datasets. Can the authors explain why?

**Relation To Broader Scientific Literature:**

- The faster optimization can improve the practical applicability of the proposed algorithm in real-world.

**Theoretical Claims:**

- Theoretical convergence of the proposed algorithm
   - While ADMM convergence is well-known for convex problems, as the authors mentioned, does the proposed ADMM with the fairness constraint also guarantee theoretical convergence?
- Question about the proof of Proposition 3.2:
   - In Eq. (24), $\inf_{V}$ is applied to $-\langle V, MH \rangle$, where it appears that $H$ is not a variable with respect to the infimum operator.
   However, in Eq. (25), $\inf_{H, V}$ is applied to $-\langle V, MH \rangle$, and in Eq. (26), $\sup_{H}$ is applied to $\langle MV, H \rangle$.
   - How can the duality (or equivalence) be proven? Or, are these results well-established?

---

> ### Author Rebuttal · Authors · 2025-03-31
>
> > Q1. FacebookNet dataset
>
> To address the reviewer's concern, we ran additional experiments on FacebookNet.
>
> **Table ztMM-1:** Clustering results on FacebookNet
> |#Clusters|Method|Time (s)|Balance|SC Cost|
> |---|---|---|---|---|
> |2|o-FSC|0.798|1.00|0.126|
> |2|s-FSC|0.193|1.00|0.126|
> |2|Ours|0.055|1.00|0.133|
> |25|o-FSC|9.908|0.84|14.114|
> |25|s-FSC|6.422|0.84|14.114|
> |25|Ours|0.131|0.84|14.128|
> |50|o-FSC|12.243|0.58|37.084|
> |50|s-FSC|11.558|0.58|37.084|
> |50|Ours|0.215|0.58|37.100|
>
> **These results further demonstrates the computational advantage of our method**, while achieving similar clustering quality to exact algorithms.
> We tested on a new large-scale dataset in *Reply to Q4 of Reviewer jPJw*.
> > Q2. ADMM convergence.
>
> Please refer to *Reply to Q1 of Reviewer 1h8o*.
> > Q3. Proof of Proposition 3.2.
>
> Our proof relies on the identification of $\phi(MH)$ as $\sup_{V} \langle V, MH \rangle - \phi^\star(V)$ which holds true as $H \mapsto \phi(MH)$ is convex. The infimum over $H$ is performed over the whole function $g(H) - \phi(MH)$, explaining how we coupled the infimums over the different variables to go from (24) to (25). We will add parentheses to better highlight that the problem we are tackling is $\inf_{H} \left [ g(H) - \sup_{V} \langle V, MH \rangle - \phi^\star(V) \right ]$.
>
> The equivalence between solving $\inf_H g(H) + \phi(MH)$ and $\inf_V \phi^\star(V) - g^\star(MH)$ is precisely the goal of Proposition 3.2. It was established in [Tonin et al. 2023] for the case of abstracts linear spaces and non-self-adjoint operators, the proof here follows a similar structure.
> > Q4. Balance interpretation.
>
> As noted in [1], Problem (6) doesn't guarantee perfect balance in general. Enforcing it may degrade spectral objective ($Tr(H^\top M H)$), so a trade-off is intrinsic to Fair SC.
>
> Our key contribution is a much faster algorithm for solving the *existing* Fair SC formulation. Fig. 2 shows our method achieves similar balance to the exact eigendecomposition, but does so substantially faster. Future work can explore regularizations to further improve balance.
> > Q5. Additional recent baselines and minimum balance.
>
> Our method follows the established line of work on fair SC, which enforces fairness via linear constraints in the embedding [1,2].
> - [Bera et al.] propose LP-based $k$-(means, median, center) clustering. Using their formulation in our work would ignore the RatioCut objective, resulting in suboptimal solutions wr.t. the spectral objective.
> - [Backurs et al.] use fairlets for prototype-based clustering.  However, extending the fairlet analysis, which relies on the $𝑘$-median and $𝑘$-center cost of the fairlet decomposition, to the spectral setting is not trivial. SC involves a spectral embedding step followed by a clustering in the embedding space, where reassigning points within a fairlet can significantly alter the spectral embedding and hence potentially violating fairness, making it non-trivial to incorporate their analysis in the fair SC case.
>
> The primary contribution of our present work is to design a significantly faster method for the already established Fair SC problem defined in [1], rather than designing alternative fair clustering problems. Therefore, these baselines are not directly comparable w.r.t. problem scope and spectral solution quality. We agree that they offer valuable alternative perspectives on fairness in the more general clustering literature and we will discuss them in our Related Works section.
>
> For balance, we follow [1,2] in reporting average balance, but also include minimum balance for Tab. 2 with $k=25$ per the reviewer’s suggestion.
>
> **Table ztMM-2:** Min balance
> |Dataset|Time (s-FSC)|Time (Ours)|Min. Balance (s-FSC)|Min. Balance (Ours)|
> |---|---|---|---|---|
> |LastFM|19.08|**4.59**|0.0027|0.0029|
> |Thyroid|30.49|**7.38**|0.0011|0.0012|
> |Census|136.60|**15.78**|0.0001|0.0001|
> |4area|166.92|**25.85**|0.1582|0.1517|
>
> > Q6. Theoretical complexity.
>
> Complexities of methods are:
> - o-FairSC: Computes null space ($O(nh^2)$) and eigendecomposition ($O((n-h)^3)$).
> - s-FairSC: Each eigensolver iteration costs $O(n^2+nh^2+nk^2)$, with the constant depending the Laplacian spectrum.
> - Ours: Dominated by matrix multiplications (applying $M$ to an $n \times k$ matrix), with $O(n^2 k)$. Since $k \ll n$, and modern libraries optimize this operation, we gain substantial practical speedups.
>
> The improvement is therefore in the efficiency of the core operations. In fact, while for an $n \times n$ matrix both matrix multiplication and matrix eigendecomposition have the same $n^3$ complexity, the former is much more efficient in practice. In the final version, we will add a separate paragraph on computational complexity detailing the above points.
>
> [1] Kleindessner et al. Guarantees for Spectral Clustering with Fairness Constraints
> [2] Wang et al. Scalable Spectral Clustering with Group Fairness Constraints

---

### Official Review · Reviewer_iKL4 · 2025-03-04

**Overall Recommendation:** 2

**Summary:**

This work addresses the issue of fairness in spectral clustering by proposing a new efficient method for fair spectral clustering (Fair SC). The authors introduce a novel algorithm that casts the Fair SC problem within the difference of convex functions (DC) framework and employs an alternating direction method of multipliers (ADMM) type of algorithm adapted to DC problems. The key contributions include a new variable augmentation strategy and the use of gradient-based algorithms to solve the subproblems efficiently, avoiding the computationally expensive eigen-decomposition required by previous methods. The paper demonstrates the effectiveness of the proposed method through numerical experiments on both synthetic and real-world datasets, showing significant speedups in computation time over prior art, especially as the problem size grows.

## Update after rebuttal

Thanks for your explanation, I will maintain my rating.

**Claims And Evidence:**

The claims made in the paper are supported by clear and convincing evidence. The authors provide a detailed derivation of the proposed algorithm and demonstrate its effectiveness through extensive experiments. The main claims are:
1. The algorithm achieves higher computational efficiency compared to existing methods (o-FSC and s-FSC).
2. The method maintains the fairness constraints while achieving comparable clustering quality.

The evidence provided includes:
1. Theoretical analysis of the DC framework and ADMM algorithm.
2. Numerical experiments on synthetic datasets (m-SBM, RandLaplace) and real-world datasets (LastFMNet, Thyroid, Census, 4area).
3. Comparison of runtime and balance metrics with existing methods.

**Essential References Not Discussed:**

There are a few essential references that could be included for a more comprehensive understanding of the context. For example, there are several new fair spectral clustering (also gradient based) methods proposed in recent years. Could you review more related work and explain the difference, and the novelty of this work?

**Experimental Designs Or Analyses:**

The experimental designs and analyses are valid. The experiments include:
- Comparison of runtime and balance metrics with existing methods (o-FSC and s-FSC).
- Sensitivity analysis of the ADMM penalty parameter α.
- Scalability analysis on datasets of varying sizes and cluster numbers.

However, it would be better to include more benchmark dataset and baseline method, like [FEN24](https://www.mdpi.com/2073-8994/17/1/12), [ZHA24](https://www.sciencedirect.com/science/article/pii/S0925231224009810?casa_token=T6cg70BHqNAAAAAA:Ow9fzmBXvJE6p3Y5vmFyUPCA-35q3KdbX2BwaSI5b1kHxBGhlxRtZp_CmyxOrokXzyvdN9oO), and other state-of-art methods for fair spectral clustering.

**Methods And Evaluation Criteria:**

The proposed methods and evaluation criteria make sense for the problem at hand. The authors use standard benchmarks and metrics to evaluate the performance of their algorithm. The evaluation criteria include: runtime, balance, clustering cost.
The methods are well-suited for the problem, as they address the computational inefficiency of previous methods while maintaining fairness constraints.

**Other Comments Or Suggestions:**

Some suggestions about writing:
1. For tables, the method with better numerical results could be highlighted.
2. Since this work is related to clustering, some figures for clustering results would be necessary to illustrate the effectiveness of the proposed method to maintain the performance and fairness.
3. Providing more details on the implementation of the proposed algorithm.
4. Could further add theory to convergence rate analysis.

**Other Strengths And Weaknesses:**

**Strength:**
1. The proposed method significantly improves the computational efficiency of Fair SC.
2. The algorithm is shown to be effective on both synthetic and real-world datasets.

**Weakness:**
1. There exists some other fair spectral clustering method, like [FEN24](https://www.mdpi.com/2073-8994/17/1/12), [ZHA24](https://www.sciencedirect.com/science/article/pii/S0925231224009810?casa_token=T6cg70BHqNAAAAAA:Ow9fzmBXvJE6p3Y5vmFyUPCA-35q3KdbX2BwaSI5b1kHxBGhlxRtZp_CmyxOrokXzyvdN9oO). [FEN24](https://www.mdpi.com/2073-8994/17/1/12) claims that they reach the same time complexity as this work. Could you add a more baseline method to this work for comparison to illustrate the strength of the proposed method.
2. When the authors switch the problem 6 (traditional fair spectral clustering optimization problem) in the work to their proposed optimization problem, they made two changes:
    - Use $M^2$ instead of $M$,
    - Enforce fairness on $MH$ instead of $H$.
Could you add some theoretical analysis to explain the feasibility of these two changes?
3. This work could review more recent related work and discuss their limitations.
4. The performance of spectral clustering and fairness looks worse than other methods.

**Questions For Authors:**

Please see weakness.

**Relation To Broader Scientific Literature:**

The key contributions of the paper are well-related to the broader scientific literature. The authors reference and build upon previous work in fair machine learning, spectral clustering, and optimization methods.

**Theoretical Claims:**

When the authors switch the problem 6 (traditional fair spectral clustering optimization problem) in the work to their proposed optimization problem, they made two changes:
1. Use $M^2$ instead of $M$,
2. Enforce fairness on $MH$ instead of $H$.

It would be better to mathematically prove the feasibility of these two changes instead of empirical results.

---

> ### Author Rebuttal · Authors · 2025-04-01
>
> > Q1. Comparative analysis with recent works [FEN24, ZHA24]
>
> We first provide discussions, followed by experiments.
> - Our work designs a much *faster* method for the *existing* Fair SC problem defined in [3]. [ZHA24] focuses on improving balance and is orthogonal to our algorithmic contribution: it learns an induced fairer graph where Fair SC is applied. Future work could combine our method on top of [ZHA24] by applying our faster algorithm to their learned graph.
> - [FEN24] does not adopt [3]. They instead introduce fairness by changing the SC objective with a different fairness regularization term; they do not employ the group-fairness constraint $F^\top H=0$. They apply coordinate descent to this new objective where they relax the discrete clustering indicator constraint _without_ orthogonality constraints $H^\top H=I$. Therefore, [FEN24] solves a different problem than (6): their solution is far from the exact one of (6) leading to higher spectral cost and potential training instability.
>
> We ran additional experiments comparing with FFSC [FEN24], following the setup in Tab. 2. **Our method is 7–12x faster and achieves significantly better spectral clustering cost and fairness constraint satisfaction.**
>
> FFSC's modified objective shows a different balance-clustering trade-off with higher balance but worse clustering and group fairness. Note the two different measures of fairness, where balance promotes same number of individuals in each cluster (Eq. (3)), and group fairness ensures proportional representation (Eq. (4)).
>
> **Table iKL4-1:** Comparison with FFSC [FEN24]
> |Dataset|Method|Time (s) (↓)|Spectral Clustering Cost (↓)|Balance (↑)|Group Fairness $\|\|F^\top H\|\|^2$ (↓)|$\|\|H^\top H - I\|\|^2$ (↓)|
> |---|---|---|---|---|---|---|
> |LastFM|FFSC|33.74|1067.906|**0.2701**|4.032|3.91E+00|
> |LastFM|Ours|**4.59**|**1.086**|0.0093|**0.000014**|**1.36E-11**|
> |Thyroid|FFSC|56.11|3080.848|**0.0170**|15.79|1.02E+01|
> |Thyroid|Ours|**7.38**|**0.353**|0.0030|**0.000001**|**2.18E-11**|
> |Census|FFSC|193.92|146669.885|**0.0051**|9.47|2.14E+00|
> |Census|Ours|**15.78**|**130.973**|0.0004|**0.000012**|**4.12E-10**|
> |4area|FFSC|237.35|285174.392|**0.6403**|58.30|2.66E+00|
> |4area|Ours|**25.85**|**242.000**|0.3823|**0.000001**|**4.22E-10**|
>
> We also ran on new datasets in *Q1 to Reviewer ztMM* and *Q4 to Reviewer jPJw*.
> > Q2. Feasibility of changes.
>
> We address the two changes:
> - Casting the Fair SC problem directly as DC using $M$ requires computing $M^{1/2}$, with complexity akin to the original problem. Standard SC seeks the top eigenvectors of $M$, which are identical to those of $M^2$. Thus, without fairness constraints, optimizing with $M^2$ yields the same solution. Extensive empirical evidence confirms that even with fairness constraints, this substitution preserves clustering quality.
> - Enforcing fairness via $MH$ instead of $H$ enables efficient dualization of the ADMM subproblem.  While in general $F^\top (MH)=0$ isn’t strictly equivalent to $F^\top H=0$, it enforces fairness on the affinity-weighted embeddings $MH$, which we noticed promotes similar groups. One intuition is that in the simpler SC case, where $H$ is top eigenvectors of $M$ as $MH = H \Lambda$ with $\Lambda$ eigenvalues, the constraint $F^\top (MH)=0$ matches $F^\top H = 0$. As shown in Tab. 3, our method achieves comparable balance and clustering cost to the exact algorithm on multiple real-world problems.
>
> While we agree that a stronger theoretical link would be ideal, we noticed that establishing it is challenging and remains an open problem.
> Extensive empirical validation supports the feasibility of this choice, making the method a strong candidate for efficient Fair SC.
> > Q3. Additional visualizations and implementation details
>
> We consider 2D datasets from [FEN24]: 'Elliptical, 'DS-577'. Figures are at https://imgur.com/a/GiBnUIs
>
> Clustering labels are shown by color; sensitive groups by shape. Legend: “C-i, G-j” for Cluster-$i$, Group-$j$.
>
> These plots show that our method produces assignments comparable to exact algorithms (o-FSC, s-FSC). Critically, we achieve this with reduced computations, as shown in the main paper. We emphasize that our main contribution lies in accelerating Fair SC - not improving fairness metrics over the existing Fair SC.
>
> Following reviewer suggestions, we will revise tables to bold best results. App. B provides implementation details, including hyperparameters, optimization settings, and $\alpha$-update rule. We will also release our code.
> > Q4. Convergence rate.
>
> For ADMM convergence, see *Q1 to Reviewer 1h8o*. Regarding rates: Deriving rates for ADMM in general nonconvex settings is challenging and typically requires assumptions difficult to verify in general for our (10). While a rate analysis is promising future work, our method shows significantly higher computational efficiency as detailed in the *Q4 to Reviewer ztMM*.
>
> [3] Kleindessner et al. Guarantees for Spectral Clustering with Fairness Constraints

---

### Official Review · Reviewer_jPJw · 2025-03-14

**Overall Recommendation:** 1

**Summary:**

This paper studies the problem of fair spectral clustering. The authors propose an ADMM-like algorithm for optimization with theoretical guarantee, and the experimental results show effectiveness in improving fairness.

**Claims And Evidence:**

Overall the claims are justified by evidences. However, the performance of s-FSC and the proposed method seems mostly comparable on all the datasets, as shown in Tab. 3. The only difference is in compuatational time, but both methods take at most few minutes. This poses question on the advantages or superiority of the proposed method.

**Essential References Not Discussed:**

I don't see any essential references missing.

**Experimental Designs Or Analyses:**

The experimental designs look sound

**Methods And Evaluation Criteria:**

The methods and criteria for comparison are limited. For example, only one recent work is considered as the alternative baseline, and only balance is considered as the fairness metric. This makes evaluating the significance of this work hard. Furthermore, most experiments are conducted on low-dimensional data, while experiments on high-dimensional and large-scale datasets are needed to validate the effectiveness.

**Other Comments Or Suggestions:**

Fig. 2 is a bit uninformative. I am not sure what conclusions I should draw from it.

**Other Strengths And Weaknesses:**

N/A.

**Questions For Authors:**

Please see above.

**Relation To Broader Scientific Literature:**

This paper can be of contribution to fair unsupervised learning. However, I am not sure if the discussion leads to a significant contribution, as the setup is limited to unsupervised clustering.

**Theoretical Claims:**

I have checked the proofs and theoretical results. However, there is no insight provided for the two propositions, and therefore it is hard to conclude theoretical contributions.

---

> ### Author Rebuttal · Authors · 2025-04-01
>
> > Q1. Our contribution within the broader literature
>
> We note that our primary contribution is to design a significantly faster method for the *existing* well-established Fair SC problem as defined in [1], rather than improving the balance of Fair SC.
>
> We respectfully disagree that the performance of our method and s-FSC is "mostly comparable".
> Our method achieving similar balance to the exact algorithm validates the quality of our found solution.
> Our method consistently improves computational efficiency over SOTA s-FSC, as shown in Tab. 2,3 and Fig. 2,3.
> The improvement is particularly evident as the sample size $n$ and number of clusters $k$ grow, as shown in Tab. 1 and Fig. 3.
> Crucially, these speed-ups stem from the fact that our algorithm replaces the eigendecomposition of s-FSC with significantly more efficient DC steps, resulting in a more scalable approach for Fair SC on real-world datasets.
>
> We want to emphasize that our work may not limit itself to “unsupervised clustering".
> Clustering is unsupervised by definition and is the main focus of this work, as stated in the Introduction.
> Our proposed DC framework exhibits favorable properties for applicability to broader settings, e.g., following work could consider constraints on the clustering solution (e.g., must-link/cannot-link constraints) to be integrated directly into the optimization problem as long as they preserve the DC structure.
> > Q2. Additional insights for the theoretical results
>
> The main contribution of this paper is designing much faster methods for spectral clustering with fairness constraints.
> This is done by reformulating the fair SC problem into a DC functions problem (Eq. (10)).
> Using this new formulation, an ADMM-type algorithm can be used to efficiently perform fair SC.
> This is because there is no need to compute the eigendecomposition of the Laplacian matrix.
>
> In short, Proposition 3.2 derives the dual of the DC formulation. This is important because existing work either relied on expensive eigendecomposition or did not consider the fairness constraints. Proposition 3.3 provides the exact closed-form expressions for all the terms involved in the dual problem so that gradient-based techniques can be applied.
> > Q3. Additional metrics and baselines
>
> To address the reviewer's concern, we have conducted additional comparisons between our method with the very recent FFSC method [FEN24] to enrich the comparative analysis.
> We also report the group-fair constraint metric (Definition 2.2). We report the results in *Table iKL4-1* in the rebuttal.
>
> We observe that **our algorithm is 7-12x faster than FFSC and results in significantly better spectral clustering cost and fairness constraint satisfaction**.
> FFSC modifies the objective with additional regularization, so it solves a different problem than ours, resulting in a different balance-clustering trade-off with higher balance but substantially worse clustering quality and group fairness metric.
> Note that these are two different measures of fairness, where balance promotes the same number of group individuals in each cluster (Eq. (3)), and the latter is related to the proportion of each group in all clusters being the same as in the general population (Eq. (4)). Additional details are given in the *Response to Q1 of Reviewer iKL4*.
>
> We additionally ran our method on the FacebookNet dataset in *Q1 to Reviewer ztMM* and also report the minimum balance metric in *Q5 to Reviewer ztMM*.
> > Q4. Large-scale experiments
>
> Our current experiments include datasets of size up to ~35,000 samples, in line with the scale of datasets commonly used in the fair SC literature, e.g., [2,FEN24].
> To address the reviewer's concern, we additionally tested on the Diabetes dataset ($n=253,680,k=2$).
> Our method results in a fair clustering in 123.36s, whereas the s-FSC (previous SOTA for [1]) did not even converge after 24 hours.
> Our method achieves fairness constraint satisfaction $||F^\top H||^2=0.000064$ (closer to 0 is better), 0.78 balance, 1.99 SC cost, and orthogonality $||H^\top H-I||^2=6.49 \times 10^{-9}$ (closer to 0 is better).
> This experiment shows that **our method allows to scale Fair SC to problems that were not even possible to solve in reasonable time** with previous algorithms.
> > Q5. Fig. 2 intuition.
>
> Fig. 2 showcases runtime and balance comparing our method and the SOTA s-FSC exact baseline.
> The left plots show that **our method consistently outperforms s-FSC** in terms of runtime, with even better efficiency gains as $k$ increases.
> The right plots compare the balance achieved by both methods, showing **our method does not decrease the fairness** compared to the exact algorithm. We will clarify this in the caption.
>
> [1] Kleindessner et al. Guarantees for Spectral Clustering with Fairness Constraints. ICML 2019
>
> [2] Wang et al. Scalable spectral clustering with group fairness constraints. AISTATS 2023
>
> [FEN24] Feng et al. Fair Spectral Clustering Based on Coordinate Descent. Symmetry 2024

---

### Official Review · Reviewer_1h8o · 2025-03-18

**Overall Recommendation:** 4

**Summary:**

This work considers fair spectral clustering, demonstrating a reformulation of the fairness constraints which allows for a considerable improvement in the runtime of existing fair algorithms. Specifically, by reformulating the trace maximization problem often used for FSC into one of multiple subproblems in the difference of convex functions setting, state of the art methods for the unfair problem are able to be employed with fairness considerations. Algorithmic ideas are sketched theoretically and further validated with experiments on real and synthetic datasets.

**Claims And Evidence:**

Claims are well supported by theoretical analysis and experimental validations. My only concerns are with respect to the convergence analysis and problem assumptions (see questions).

**Essential References Not Discussed:**

n/a

**Experimental Designs Or Analyses:**

The experimental design appears valid / standard for the problem at hand.

**Methods And Evaluation Criteria:**

Yes.

**Other Comments Or Suggestions:**

n/a

**Other Strengths And Weaknesses:**

Strengths:
The paper is well written and provides a nice introduction to spectral clustering + the methods leveraged to ensure added fairness constraints are met. Moreover, the results (experimental in particular) are compelling to demonstrate a significant computational improvement over the prior state of the art.

Weaknesses:
The noted assumption that M is full rank seems potentially weak in practice. For example, the authors note that this is violated when datasets have duplicates--a very common issue when collecting data. Correct me if I'm wrong on this!

**Questions For Authors:**

Can you explain why the balance is worse for your algorithm for small values of k?

Is the assumption that full rank assumption on M standard? In what contexts might this not hold / how much does it weaken the presented results?

Can you expand on the convergence guarantee of this algorithm?

Can this difference of convex function reframing be used to capture other notions of fairness (ie. not just the notion of balance)?

**Relation To Broader Scientific Literature:**

The authors effectively contrast their results against prior work in spectral clustering (with and without fairness constraints). As a non-expert on this problem, the paper caught me up to speed well.

**Theoretical Claims:**

I skimmed the proofs deferred to the appendix and did not note any major issues. I do feel the paper lacks a more thorough theoretical treatment of the convergence guarantees (discussed in Section 3.2). Moreover some of the problem assumptions are not immediately clear to me (see my questions).

---

> ### Author Rebuttal · Authors · 2025-03-31
>
> We thank the reviewer for the valuable feedback and address the remaining concerns below.
>
> > Q1. Expand on the convergence analysis.
>
> Our convergence analysis is based on the theory for ADMM applied to nonconvex problems from [1], which analyzes the convergence of ADMM for structured nonconvex problems of the form presented in our Equation (11) (a composite function minimization with linear constraints).
>
> Applying Proposition 3 from [1] to our specific ADMM structure, we can assert the following: Provided the sequence of dual variables ($P^{(i)}$) generated by the algorithm converges (i.e., $\lim_{i\to \infty} P^{(i)} = P^\star$ for some $P^\star$), and under mild assumptions (Assumptions 7, 8, 9 in [1], which correspond to our problem having closed sets, local solutions being found for subproblems, and regularity assumptions), then any limit point ($H^\star, Y^\star$) of the primal sequence ($H^{(i)}, Y^{(i)}$) satisfies first-order conditions.
>
> We will revise the paragraph on "Convergence analysis" on page 6 to more formally state in a separate Proposition the convergence guarantee and its conditional nature on dual convergence based on [1], due to the technical challenges in this nonconvex setting.
>
> > Q2. Balance interepretation as $k$ changes.
>
> Fig. 2 shows that balance typically decreases as the number of clusters $k$ increases.
> This trend is not unique to our algorithm.
> It is observed in both our method (orange line) and the exact s-FSC algorithm (blue line).
> As $k$ increases, the expected size of each cluster diminishes.
> Smaller clusters can lead to a greater number of highly unbalanced groups.
> The reported balance metric is the average of (3), i.e., the minimum ratio of group ratios in each single cluster [2].
> Consequently, even if a few clusters exhibit poor balance (due to small cluster size), this can significantly lower the balance score.
> Similar trends and reasonings can be found in [2, Sec. 6.1].
> From a theoretical angle, [2, Theorem 1] notes that, if $k$ increases for fixed $n$, the fairness recovery of fair SC (in the SBM setting) becomes looser.
> We also note that the primary contribution of our present work is to design a significantly faster method for the _existing_ Fair SC problem, rather than improving the balance of Fair SC.
>
> > Q3. Rank of $M$.
>
> - This is a technical assumption needed to compute the gradient of $g^\star$ from Proposition 3.3.
> - Regarding duplicates in the data. While in regression it may be meaningful to have duplicated data as the same input can result in a different output (e.g., a different measurement), in unsupervised learning the duplicates just lead to a simple re-weighting of the empirical risk. Therefore, one can simply filter the duplicates in a pre-processing step.
> - This assumption can be satisfied using infinite-dimensional kernels $\kappa$. Specifically, this is always the case with universal kernels (e.g., the Gaussian kernel) [4].
> - In case the given data leads to $M$ not being full rank, the problem can be regularized with $\mathcal{M} := M+(1+\omega)I$ for small $\omega>0$. This is a well-studied regularization technique that is widely used, e.g., in kernel methods [3].
>
> > Q4. DC applicability to other settings.
>
> The reviewer raises an interesting point regarding the applicability of our framework to more general setting/notions.
>
> For example, it might be possible to extend it to constrained spectral clustering.
> By considering a modified indicator function $h(\cdot)$, our framework can accommodate other constraints/fairness metrics whenever the DC formulation is maintained. The ADMM algorithm (Algorithm 1) can then be applied with appropriate modifications to the subproblem w.r.t. $Y$. For instance, linear constraints have been used in the literature to encode must-link and cannot-link constraints within spectral clustering [5].
>
> We will add a remark in the final version of the paper to reflect this potential broader application of our method.
>
> ---
>
> [1] Magnússon et al. "On the convergence of alternating direction lagrangian methods for nonconvex structured optimization problems." IEEE Transactions on Control of Network Systems (2015).
>
> [2] Chierichetti et al. Fair Clustering Through Fairlets. NeurIPS 2017.
>
> [3] Christopher M Bishop and Nasser M Nasrabadi. Pattern recognition and machine learning. Springer, 2006
>
> [4] Ingo Steinwart and Andreas Christmann. Support vector machines. Springer Science & Business Media, 2008.
>
> [5] Kawale et al. "Constrained spectral clustering using l1 regularization." International Conference on Data Mining (2013).

---

### Decision · Program_Chairs · 2025-05-01

**Decision:**

Accept (poster)

**Comment:**

This paper presents a new take on fair spectral clustering, with a focus on runtime/scalability.  The theory is solid and the experimental results are strong but can be expanded (see following).  This AC is broadly discounting reviewer jPJw’s very negative score because (i) some of the complaints just miss the point, e.g., Table 3 shows a baseline s-FSC compared against the proposed method with similar objective scores, but dramatically improved runtimes - that’s the point - and (ii) a lack of response to what I view as a comprehensive and convincing rebuttal on the authors’ part.  This AC does strongly suggest that the authors make the additional changes that iKL4, jPJw, and others bring up - that there are other spectral methods that would provide good baselines, and to compare against them, especially the ones with similar asymptotic runtimes as mentioned in iKL4’s review.  It’s great the authors already ran against FFSC in the rebuttal, with the same runtime lifts as in Table 3 against s-FSC, but for longer-term impact it’s always good to compare against the full slate of methods.